# Ordered patterning of the sensory system is susceptible to stochastic features of gene expression

**Ritika Giri[1,2], Dimitrios K Papadopoulos[3†], Diana M Posadas[1], Hemanth K Potluri[1], Pavel Tomancak[3], Madhav Mani[1,2,4]\*, Richard W Carthew[1,2]\***

[1]Department of Molecular Biosciences, Northwestern University, Evanston, United States; [2]NSF-Simons Center for Quantitative Biology, Northwestern University, Evanston, United States; [3]Max Planck Institute of Cell Biology and Genetics, Dresden, Germany; [4]Department of Engineering Sciences and Applied Mathematics, Northwestern University, Evanston, United States

**\*For correspondence:**
madhav.mani@northwestern.edu
(MM);
r-carthew@northwestern.edu
(RWC)

**Present address:** †MRC Institute of Genetics and Molecular Medicine, University of Edinburgh, Edinburgh, United Kingdom

**Competing interests:** The authors declare that no competing interests exist.

**Abstract** Sensory neuron numbers and positions are precisely organized to accurately map environmental signals in the brain. This precision emerges from biochemical processes within and between cells that are inherently stochastic. We investigated impact of stochastic gene expression on pattern formation, focusing on *senseless* (*sens*), a key determinant of sensory fate in *Drosophila*. Perturbing microRNA regulation or genomic location of *sens* produced distinct noise signatures. Noise was greatly enhanced when both *sens* alleles were present in homologous loci such that each allele was regulated in trans by the other allele. This led to disordered patterning. In contrast, loss of microRNA repression of *sens* increased protein abundance but not sensory pattern disorder. This suggests that gene expression stochasticity is a critical feature that must be constrained during development to allow rapid yet accurate cell fate resolution.

## Introduction

The irreversible progression from a disordered to an ordered arrangement of cells within tissues is a hallmark of development. Developing organisms rely on precise control of cellular gene expression in order to achieve this outcome. However, biochemical reactions such as transcription and translation involve stochastic molecular collisions subject to intrinsic variability (*Blake et al., 2003*; *Cai et al., 2006*; *Elowitz et al., 2002*; *Newman et al., 2006*; *Ozbudak et al., 2002*; *Taniguchi et al., 2010*). Therefore, a central question in developmental biology concerns how probabilistic gene expression generates deterministic developmental outcomes.

Fluctuations in mRNA and protein numbers occur because of random birth and death of these molecules (*Paulsson, 2005*; *Thattai and van Oudenaarden, 2001*). Since one molecule of mRNA is usually translated into multiple copies of proteins, small fluctuations in mRNA number can lead to larger fluctuations in protein number (*Elowitz et al., 2002*; *Ozbudak et al., 2002*; *Paulsson, 2005*; *Thattai and van Oudenaarden, 2001*). In theory, the stochasticity in protein copy number caused by birth-death processes will be mitigated if large numbers of protein molecules are present in each cell (*Seneta, 2013*). Indeed, many transcription factors are reported to be expressed in excess of $10^4$–$10^5$ protein copies in terminally fated cells (*Biggin, 2011*). However, it is unclear how many copies of such fate-determining proteins are present at cell-fate decision points, and therefore how extensively the stochasticity inherent to birth-death processes impinges upon fate decisions.

There are additional sources of noise in gene expression. Many genes are transcribed in bursts (*Bothma et al., 2014*; *Chubb et al., 2006*; *Dar et al., 2012*; *Garcia et al., 2013*; *Golding et al., 2005*; *Raj et al., 2006*; *Rodriguez et al., 2019*; *Suter et al., 2011*). Such genes switch stochastically

between an actively transcribing state and an inactive non-transcribing state. This generates bursts of newly synthesized mRNA molecules interspersed with periods of dormancy. Various physical features of gene promoters, their enhancers, and the transcription factors that bind to them have been shown to affect the burstiness of gene transcription (*Jones et al., 2014*; *Sanchez and Golding, 2013*). Several mechanisms have been proposed to buffer protein numbers against bursty mRNA fluctuations. These include spatial and temporal averaging of transcript numbers (*Bahar Halpern et al., 2015*; *Garcia et al., 2013*; *Gregor et al., 2007*; *Raj et al., 2006*), polymerase pausing (*Boettiger and Levine, 2009*), and autoregulation of gene expression (*Boettiger and Levine, 2013*; *Papadopoulos et al., 2019*).

Since tissues are patterned by the actions of gene regulatory networks (GRNs) across diverse temporal and spatial-scales, efforts are being made to understand how stochastic expression of these genes affects pattern formation (*Boettiger and Levine, 2013*; *Bothma et al., 2015*; *Gregor et al., 2007*; *Raj et al., 2010*; *Tkacik et al., 2008*; *Zoller et al., 2018*). We have focused on a patterning system involving the Wnt and Notch signaling pathways, which are two widespread means for cells to communicate with one another (*Hayward et al., 2008*; *Pires-daSilva and Sommer, 2003*).

Often, Wnt and Notch signals intersect upon a set of cells, and from this emerge precise patterns of differentiated cells (*Collu et al., 2014*; *van Es et al., 2005*; *Hayward et al., 2008*; *Peter and Davidson, 2011*; *Sonnen et al., 2018*). A classic example of such an intersection is the emergence of rows of sensory organ (S) cells located alongside the dorsal and ventral (DV) compartment boundary of the *Drosophila* wing imaginal disc (*Figure 1A*). Each row of S fated cells develops into a highly ordered row of sensory bristles located at the anterior margin of the adult wing (*Figure 1B*). DV boundary cells in the wing disc secrete the Wnt ligand Wingless (Wg) (*Couso et al., 1993*; *Zecca et al., 1996*), which induces stripes of nearby cells to express proneural genes including *senseless* (*sens*) (*Eivers et al., 2009*; *Jafar-Nejad et al., 2006*; *Phillips and Whittle, 1993*; *Figure 1C*).

Each proneural stripe then self-organizes into a periodic pattern of high and low Sens expressing cells (*Figure 1A*). This is orchestrated by two counteracting regulatory loops. The proneural proteins are transcription factors that proportionally stimulate expression of the Notch ligand Delta (*Hinz et al., 1994*; *Nolo et al., 2001*). Delta activates Notch in neighboring cells and thereby inhibits proneural gene expression in these cells. This generates classic lateral inhibition. At the same time, the proneural proteins co-activate their own transcription within each cell (*Acar et al., 2006*; *Jafar-Nejad et al., 2003*; *Jafar-Nejad et al., 2006*; *Nolo et al., 2000*). These interlinked positive feedback loops ensure that initially small differences in proneural protein abundance between neighboring cells evolve into large differences (*Figure 1C*). While sustained and strong expression of Sens is sufficient to drive a cell towards the S fate, neighboring cells downregulate Sens and adopt an epidermal (E) fate (*Jafar-Nejad et al., 2003*).

Since lateral inhibition harnesses the variation in proneural protein abundance, we have sought to understand if stochasticity in proneural gene expression is filtered out by spatial signal integration between cells; or transmitted across scales to disrupt ordered sensory bristle patterning. We have measured the stochastic properties of Sens protein expression and have used experimental perturbations and mathematical modeling to determine the sources of noise. As anticipated, we discover that molecular birth-death processes and transcriptional bursting influence the stochastic features of Sens expression. Surprisingly, we find that stochastic features of Sens protein expression are greatly enhanced when one *sens* allele influences the expression of its paired homolog in trans. When this occurs, cells in the proneural stripes experience abnormally high noise in Sens protein output, which resolves by lateral inhibition into a disordered pattern of sensory bristles. Thus, *cis* versus *trans* modes of gene regulation can have major effects on the regularity of sensory pattern formation.

## Results

### Counting sens proteins to measure expression noise

Protein fluctuations can be observed by counting molecules in single cells over time (*Figure 1D*). Alternatively, noise can be estimated by tagging the two alleles of a gene with distinct fluorescent proteins and measuring fluorescence correlation in individual cells (*Figure 1E*; *Elowitz et al., 2002*;

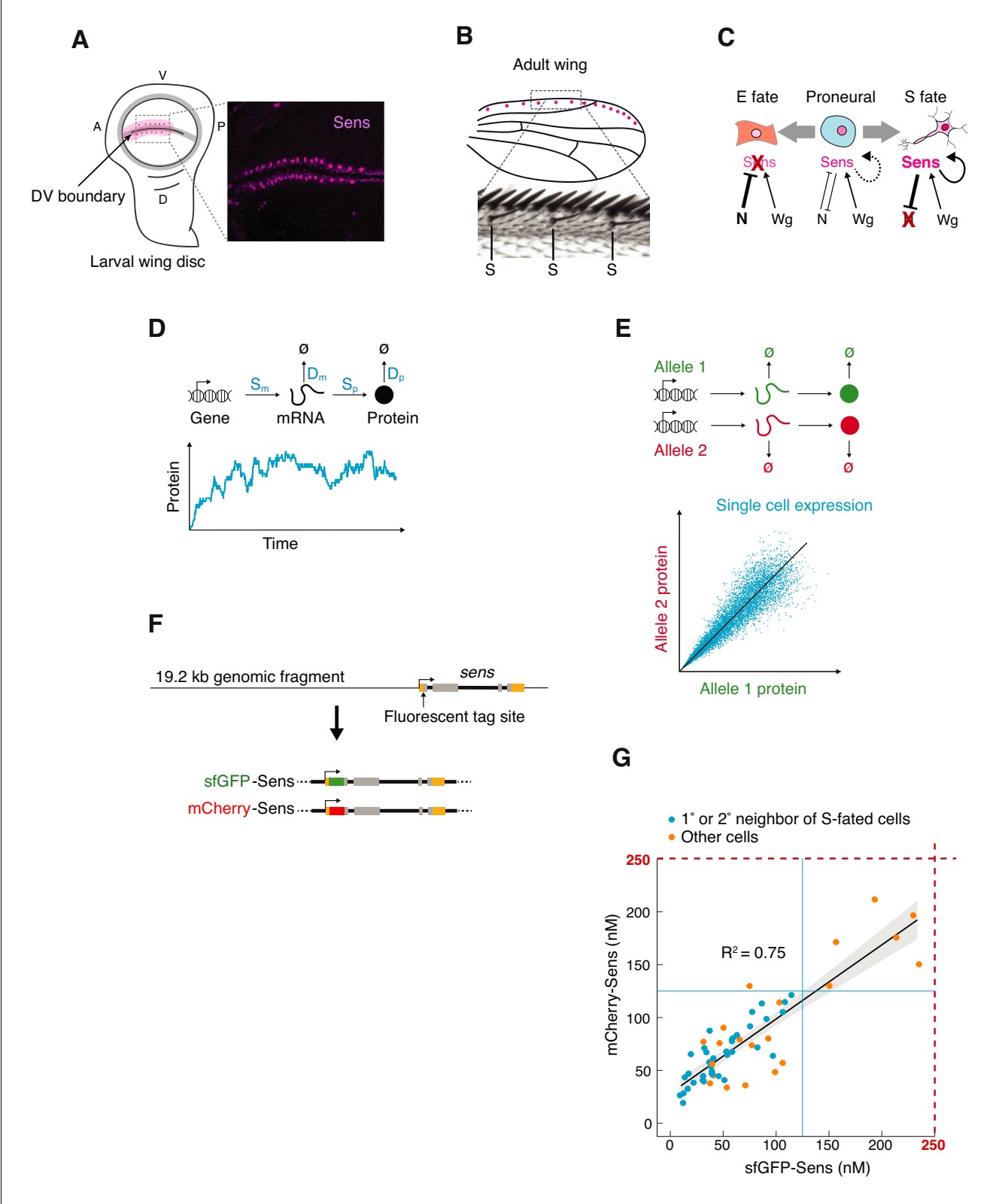

**Figure 1.** Measuring *sens* gene expression stochasticity during sensory organ fate selection. (**A**) Sens protein is expressed in two stripes of cells bordering the dorsoventral (DV) boundary of the wing disc. The pattern refines into a periodic pattern of S-fated cells in the anterior region, which can be seen as expressing high levels of Sens protein. Anterior (A), left. Ventral (V), top. Right panel is a micrograph of Sens protein immunofluorescence. (**B**) This generates the highly ordered pattern of sensory bristles along the anterior margin of the adult wing. S denotes chemosensory bristles that had

*Figure 1 continued on next page*

*Figure 1 continued*

been determined at the stage visualized in (**A**). (**C**) Cells are induced by Wg to a proneural state expressing moderate levels of Sens. Notch-mediated lateral inhibition causes cells to switch to either low stable expression (E fate) or high stable expression (S fate) of Sens. Cell-autonomous positive feedback by Sens and non-autonomous feedback by mutual inhibition are key to this process. (**D**) Gene expression output is inherently variable due to stochastic synthesis and decay of mRNA and protein molecules. Therefore, single cell protein counts fluctuate stochastically around the expected steady state expression level. The magnitude of these fluctuations is determined by the rate constants of individual steps (in blue). (**E**) Stochasticity can be measured by tagging the two alleles of a gene with distinct fluorescent proteins and measuring fluorescence correlation in individual cells. Each datapoint is red and green fluorescence in one cell. Cells with greater gene expression stochasticity deviate further from the expected average fluorescence (black line). (**F**) A genomic fragment containing *sens* was N-terminally tagged with either single sfGFP or mCherry tags. These were used to rescue *sens* mutant animals by site-specific insertion into genomic location 22A3 (*Figure 1—figure supplement 1*). (**G**) Single-cell mCherry and sfGFP protein numbers counted by FCS in *sfGFP-sens/mCherry sens* wing cells (see also *Figure 1—figure supplements 2–3*).

The online version of this article includes the following figure supplement(s) for figure 1:

**Figure supplement 1.** The *sens* transgene inserted at 22A3 expresses Sens similarly to the endogenous *sens* gene and it rescues all mutant phenotypes.
**Figure supplement 2.** Image analysis of developing wing tissue to quantify Sens protein in single cells.
**Figure supplement 3.** Calibrating a conversion factor between confocal microscopy fluorescence and FCS molecule counting.

*Raser and O'Shea, 2004*). Protein number from each allele is stochastically fluctuating over time, and their fluctuations are independent of one another. When one fixes a population of cells and measures protein output from each allele per cell, the limited correlation between allele output for the population closely approximates the stochastic variability if one were to measure noise temporally (*Swain et al., 2002*). We created *sens* alleles tagged with superfolder GFP (sfGFP) or mCherry (*Figure 1F*). This was done by modifying a 19.2 kb fragment of the *Drosophila* genome containing *sens* by singly inserting either sfGFP or mCherry into the amino terminus of the *sens* ORF (*Figure 1F*). These transgenes were independently landed into the 22A3 locus on the second chromosome, a standard landing site for transgenes (*Venken et al., 2006*; *Venken et al., 2009*). Endogenous *sens* activity was inhibited with amorphic loss-of-function mutations (*Jafar-Nejad et al., 2003*; *Nolo et al., 2000*). The transgenes completely rescued all detectable *sens* mutant phenotypes and exhibited normal expression (*Figure 1—figure supplement 1*). We then mated singly-tagged *sfGFP-sens* with *mCherry-sens* animals to generate trans-heterozygous progeny in an endogenous *sens* null background. Wing imaginal discs of these offspring were fixed and imaged by confocal microscopy (*Figure 1—figure supplement 2A*). Cells were computationally segmented in order to measure intensity of sfGFP and mCherry fluorescence within each cell (*Figure 1—figure supplement 2*).

Relative fluorescence units (RFU) were converted into absolute numbers of Sens protein molecules via Fluorescence Correlation Spectroscopy (FCS), which measured the absolute concentrations of sfGFP-Sens and mCherry-Sens protein in live wing discs (*Figure 1G*). For both sfGFP and mCherry FCS measurements, the highest Sens levels were no greater than 250 nM (*Figure 1G*), corresponding to approximately 25 RFU obtained from imaging (*Figure 1—figure supplement 3A*). This gives an approximate conversion factor of 1 RFU equivalent to 10 nM. To validate this estimate, we classified Sens-positive cells and mapped them onto the raw images (*Figure 1—figure supplement 3C*). An accurate estimate of a conversion factor would recreate the expected expression pattern of Sens, where S-fated cells are periodically dispersed along both sides of the DV boundary surrounded by first and second order neighbors (*Figure 1—figure supplement 3B*). We reproducibly observed this pattern for a conversion factor of 10 but not when it was varied three-fold lower or higher (*Figure 1—figure supplement 3C*). Since the average wing disc cell nuclear volume is $23 \times 10^{-15}$ L (*Papadopoulos et al., 2019*), 1 RFU is estimated to correspond to ~138 Sens molecules.

## Sens protein noise displays a signature arising from birth-death processes

Applying the conversion factor, we observed that wing disc cells displayed a broad range of Sens protein molecule numbers (*Figure 2A*). This observation is consistent with earlier studies using anti-Sens immunofluorescence (*Jafar-Nejad et al., 2006*). Although both sfGFP and mCherry tagged alleles contributed equally to total Sens protein output on average (*Figure 1G*), there were significant intracellular differences between sfGFP-Sens and mCherry-Sens molecule numbers (*Figure 2A*).

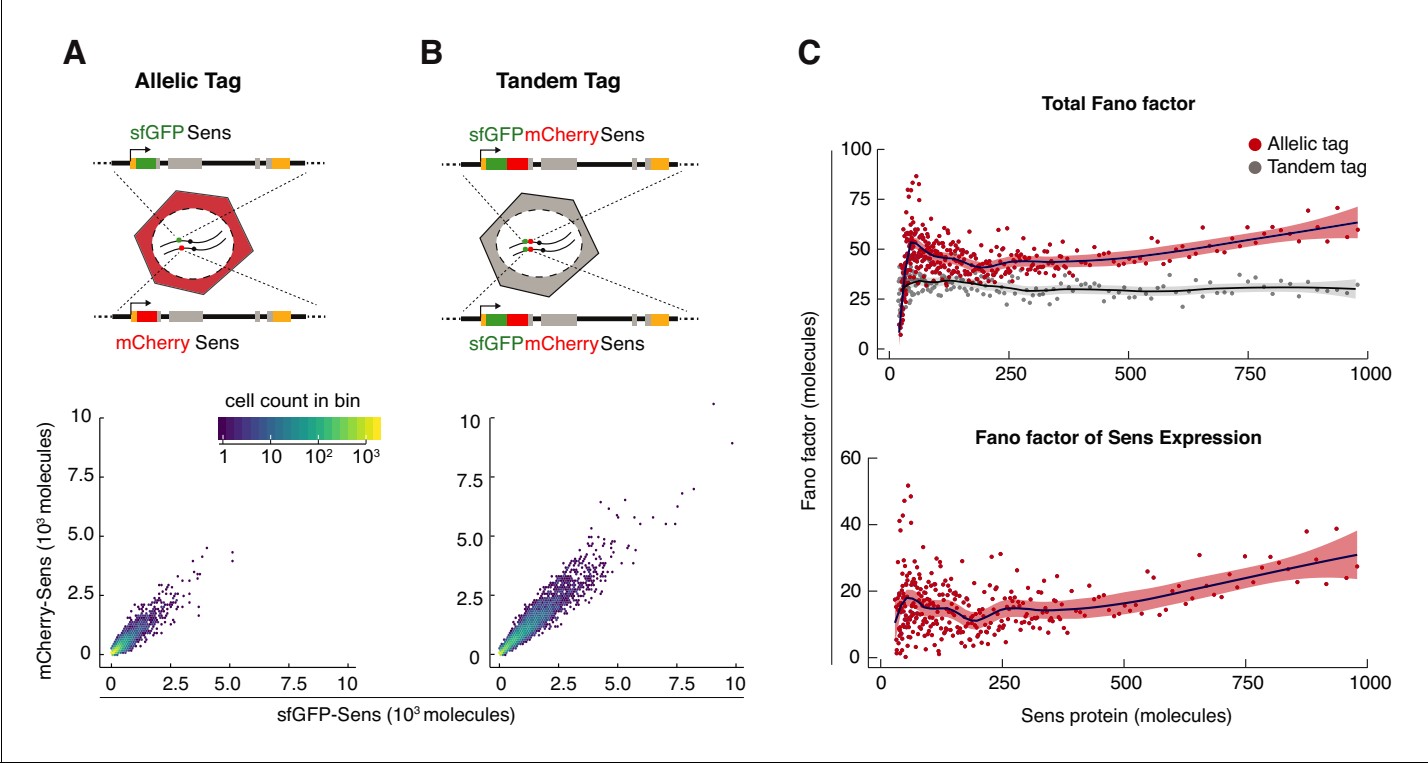

**Figure 2.** Sens Fano factor relative to protein copy number in single cells. (A) sfGFP- and mCherry-Sens protein numbers measured from single cells in *sfGFP-sens/mCherry-sens* wing discs. 12,000 cells were plotted for comparison in (A,B). Colors represent cell count in each hexagonal plot region. (B) Estimated sfGFP and mCherry numbers generated from the tandem-tagged *sfGFP-mCherry-sens* gene in single cells of wing discs. Note that these numbers are not perfectly correlated with one another due to noise in the measurement process (see also *Figure 2—figure supplement 1*). As expected, median expression of the tandem tag was 2.1 ± 0.03 fold higher than allelic tag expression. (C) Top panel: The Fano factor was calculated in bins of cells expressing either tandem-tagged Sens or the singly-tagged allelic pairs of Sens (*Figure 2—figure supplement 2*). Bottom panel: The Fano factor of Sens expression was calculated by subtracting out the Fano factor from tandem-tagged cells. Plotted region encompasses data from 88% of tagged cells (Sens < 1000 molecules). Shading demarcates 95% confidence intervals.

The online version of this article includes the following figure supplement(s) for figure 2:

**Figure supplement 1.** Fluorescence Resonance Energy Transfer (FRET) from sfGFP to mCherry molecules is negligible under experimental imaging conditions.

**Figure supplement 2.** Fano factor calculation from nuclear fluorescence signals of Sens positive cells.

This was due to two independent sources of noise: (1) stochastic gene expression, (2) stochastic processes in the measurement of protein number. The latter source of noise arises from differential rates of protein folding and turnover, probabilistic photon emission and detection, as well as image analysis errors. To estimate measurement noise, we constructed a third transgene containing both sfGFP and mCherry fused in tandem to the *sens* ORF (*Figure 2B*). *sfGFP-mCherry-sens* was inserted at locus 22A3, and fluorescence was measured in disc cells from such animals. Since sfGFP and mCherry molecule numbers should be perfectly correlated when expressed as a tandem-tagged protein in vivo, we attributed any decrease in fluorescence correlation to measurement noise (*Figure 2B*). Negligible Fluorescence Resonance Energy Transfer (FRET) was observed between mCherry and sfGFP proteins in tandem-tagged *sfGFP-mCherry-sens* cells, indicating that stochastic noise was not under-estimated due to FRET interactions (*Figure 2—figure supplement 1*).

Intrinsic noise of protein expression $\eta^2$ has been defined as the extent to which protein output from two alleles of a gene fail to correlate in the same cell (*Elowitz et al., 2002*). The value of $\eta^2$ indicates the mean relative difference between the two reporter proteins in the same cell; for instance, a value of 0.10 would indicate the two reporter proteins differ by about 10% of the average expression. Since we had quantified the absolute number of protein molecules, we expressed noise as the Fano factor, which is defined as

$$Fano\ factor = \eta^2 . \mu$$

where μ is the mean protein number. This factor indicates the average difference in number of molecules between the two reporter proteins at any given time.

A benefit of calculating the Fano factor is to determine if the protein noise can be modeled as a Poisson-like process. Previous studies using dissociated cells have shown that protein noise, expressed as the Fano factor, remains constant as protein output varies (*Bar-Even et al., 2006*; *Elowitz et al., 2002*). This is due to stochastic birth and death of mRNA and protein molecules (*Paulsson, 2005*; *Thattai and van Oudenaarden, 2001*). We estimated the empirical Fano factor as a function of Sens protein output in cells expressing either singly-tagged Sens or tandem-tagged Sens (*Figure 2C* and *Figure 2—figure supplement 2*). To estimate the Fano factor due to stochasticity of Sens expression, we subtracted out the technical contribution as measured in tandem-tagged cells. The Fano factor for Sens expression displayed a complex relationship to protein output, with a minor peak in cells containing fewer than 200 molecules, and then dropping to a level that slowly rose with higher Sens output (*Figure 2C*).

To understand the origins of this profile, we created a simplified model of gene expression (*Figure 3A*). Each reaction in the model was treated as a probabilistic event, reflecting the stochastic nature of gene expression (*Figure 3—figure supplement 1A–C*; *Paulsson, 2005*; *Thattai and van Oudenaarden, 2001*). Using rate parameters that we measured for *sens* expression in the wing (*Figure 3—figure supplement 2* and *Figure 3—source data 1*), we ran thousands of simulations mimicking protein output from two independent alleles in each virtual cell (*Figure 3—figure supplement 1B,C*).

We first considered a model in which the promoter was always active (*Figure 3A*). The simulated Fano factor was constant irrespective of protein number, consistent with the noise being caused by random birth-death events. This somewhat resembled the Sens profile of Fano factor experimentally observed in cells containing more than 200 molecules of Sens (*Figure 2C*). However, there was a weak rise in the observed Fano factor, and so we hypothesize that one of the post-transcriptional rate constants weakly varies as a function of protein output (*Figure 3—figure supplement 1D*).

## Experimental validation of model prediction on birth-death processes and noise

The model predicts that mRNA and protein birth-death processes contribute a uniform level of noise, expressed as the Fano factor, across the entire spectrum of Sens output. Note that the experimentally measured Fano factor is the cumulative sum of Fano noise from each allele. Therefore, expression stochasticity from one allele contributes to half the cumulative sum. This value of the Fano factor is predicted to be equivalent to the average number of protein molecules translated from one mRNA molecule in its lifetime, also called the translation burst size (*Paulsson, 2005*; *Schmiedel et al., 2015*; *Thattai and van Oudenaarden, 2001*). To test this model prediction, we experimentally altered the translation burst size for Sens. We did so by eliminating the post-transcriptional repression of *sens* by the microRNA miR-9a (*Cassidy et al., 2013*; *Li et al., 2006*). Since microRNAs inhibit translation output and/or mRNA lifetime, loss of microRNA-mediated repression should increase the translation burst size of a target gene proportional to the magnitude of de-repression of the target.

We eliminated miR-9a regulation of *sens* by mutation of the two binding sites for miR-9a in the *sens* mRNA 3'UTR. This abolishes the impact of miR-9a on *sens* gene expression (*Cassidy et al., 2013*). We then estimated the fold-increase in Sens protein number when miR-9a regulation is lost. We did so by comparing Sens output from *sens* alleles with intact or mutated miR-9a sites (*Figure 4A*). Loss of miR-9a repression increased Sens protein number an average of 1.8-fold in all cells (*Figure 4B,C*). Similar fold-repression values were observed irrespective of the fluorescent tag used to estimate the ratio of mutant-to-wildtype Sens molecules (*Figure 4—figure supplement 1*). If Sens noise arises from birth-death processes, we predicted that the Fano factor would uniformly increase by ~1.8 fold for the mutant *sens* gene, reflecting its greater translation burst size. We measured the Fano factor for cells where both *sens* alleles contained mutated miR-9a sites (*Figure 4D*). Loss of miR-9a regulation increased the Fano factor uniformly across the entire range of Sens protein

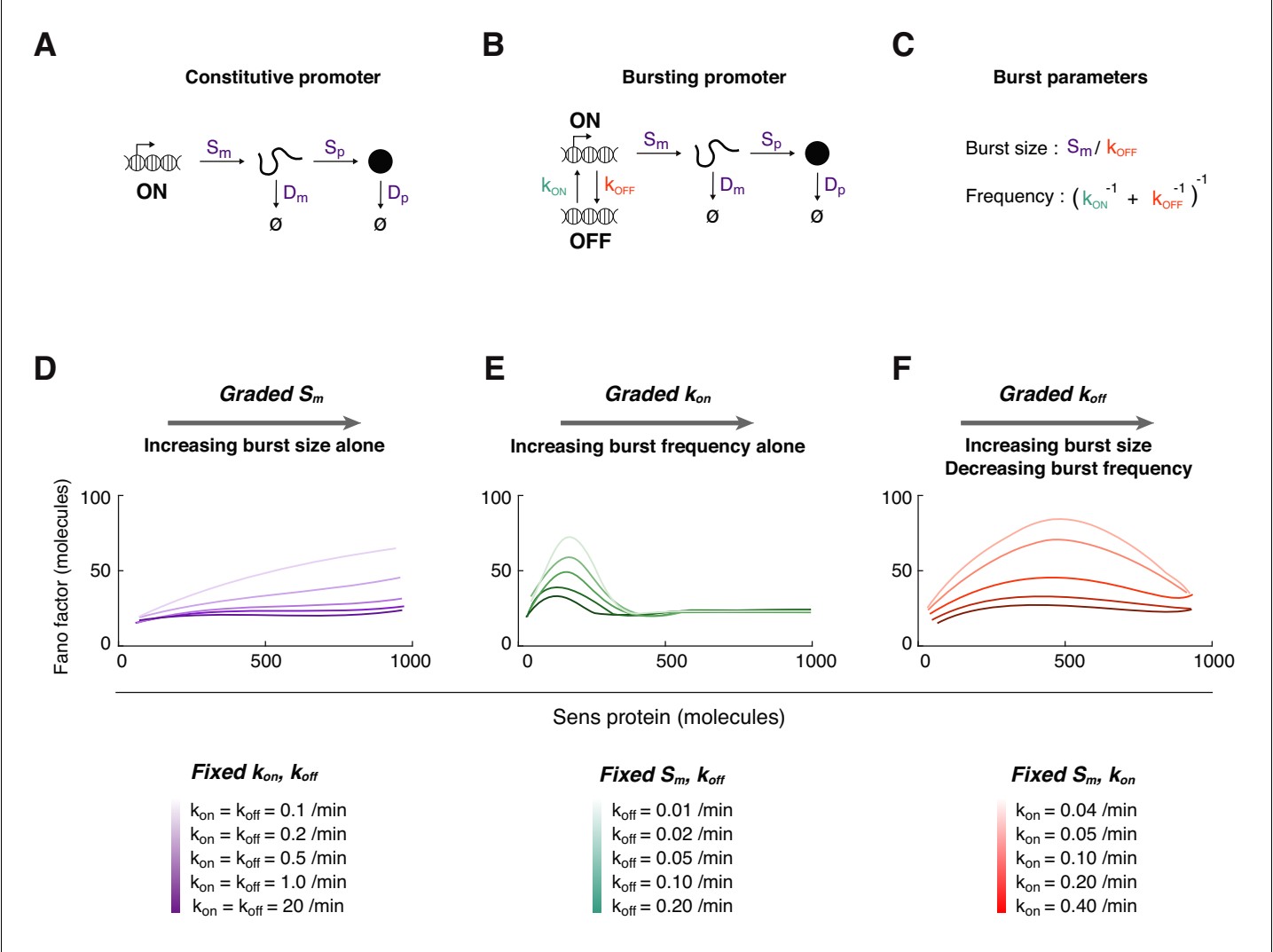

**Figure 3.** Mathematical modeling of Sens protein noise. (A) The model of gene expression with a constitutively active promoter. (B) The two-state model with a promoter having distinct on and off states, and independent rate constants for state conversion (see also *Figure 3—figure supplements 1–2*). (C) The relation between transcription rate constants and transcription burst frequency and burst size. (D-F) The Fano factor derived from simulations of the two-state model. In each panel, a different transcription rate constant is varied to generate a range of Sens protein output. Trend lines were generated by smoothing the Fano factor profiles obtained from binning 5,000 simulated cells. (D) The rate constant $S_m$ is systematically varied from 0.1 to 1 mRNA/min. Rate constants $k_{on}$ and $k_{off}$ are fixed at the values shown for each simulation curve. The promoter switches rapidly at high values of $k_{on}$ and $k_{off}$ such that the Fano factor is similar to one from a constitutively active promoter (dark purple line). (E) The rate constant $k_{on}$ is systematically varied from 0.025 to 10 min$^{-1}$. Rate constant $S_m$ is fixed at 0.25 mRNA/min, and $k_{off}$ is fixed at the values shown for each simulation curve. (F) The rate constant $k_{off}$ is systematically varied from 0.025 to 3 min$^{-1}$. Rate constant $S_m$ is fixed at 0.5 mRNA/min, and $k_{on}$ is fixed at the values shown for each simulation curve.

The online version of this article includes the following source data and figure supplement(s) for figure 3:

**Source data 1.** Rate parameters used in modeling.
**Figure supplement 1.** Modeling of Sens expression and noise.
**Figure supplement 2.** Measurement of *sens* mRNA and protein decay in the wing disc.

output. The increase in overall Fano factor was approximately two-fold. This result is consistent with stochastic birth-death processes uniformly contributing to Sens protein noise.

## Sens protein noise displays a signature arising from transcription bursts

Strikingly, when miR-9 repression was lost, the small peak in Fano factor was strongly enhanced in cells with 300 molecules/cell or less (*Figure 4D*). The existence of this peak was not predicted by

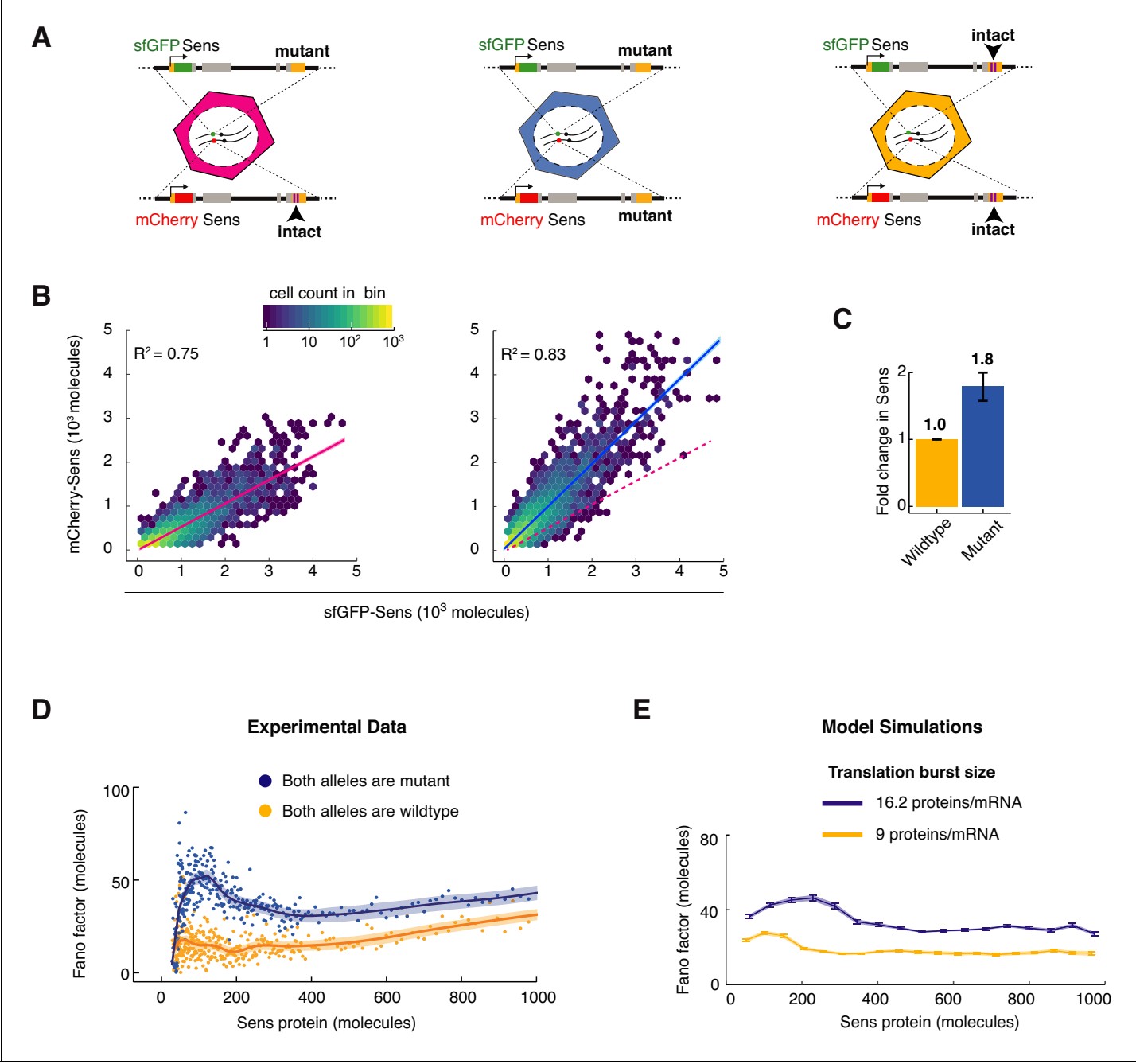

**Figure 4.** MicroRNA regulation uniformly decreases Sens protein output and noise. (**A**) The *sens* transgenes were modified to mutate the two miR-9a binding sites in the 3'UTR. Wing disc cells were generated that had different combinations of mutant and wildtype alleles. (**B**) Protein output measured from the *mCherry-sens* allele with intact (left plot) or mutated (right plot) miR-9a sites. These were measured relative to protein output from the *sfGFP-sens* allele with mutated sites. A sample of 6,000 cells is shown for each plot, and colors represent cell density in the plotted hexagonal bins. Lines represent the linear regression model fit and shaded regions are the 95% confidence intervals. Dotted red line in right plot shows fit from the left plot. Since the linear model fits both genotypes equally well, it argues that the strength of miR-9a repression is equivalent in all cells, regardless of Sens output. (**C**) Sens protein output increases with loss of miR-9a binding sites. Error bars are standard error of the mean. Shown is the comparison of mutant *sfGFP-sens* to wildtype *mCherry-sens* but the same result was observed if the mutant *mCherry-sens* was compared to wildtype *sfGFP-sens* (*Figure 4—figure supplement 1*). (**D**) Loss of miR-9a regulation leads to an increase in Fano factor across the entire range of Sens protein output. Shaded regions are 95% confidence intervals. (**E**) Model simulations with a 1.8-fold increase in translation burst size (defined as *Sp/Dm*) reproduces the experimentally observed Fano profile. Error bars are 95% confidence intervals.

The online version of this article includes the following figure supplement(s) for figure 4:

**Figure supplement 1.** Fluorescent protein tags sfGFP and mCherry behave interchangeably in vivo.

stochastic birth-death processes alone. Therefore, we considered a more complex model of gene expression (*Figure 3B*). The promoter was allowed to switch between active and inactive states such that it transcribed mRNA molecules in bursts (*Figure 3C*). When we systematically varied the promoter activation parameter $k_{on}$, the in-silico Fano factor profile exhibited a peak when protein output was low (*Figure 3E*). This trend was seen when the other parameters were fixed at different values, with the amplitude and position of the peak changing but always biased to lower protein output. In contrast, varying the initiation parameter $S_m$ or inactivation parameter $k_{off}$ did not yield a Fano peak when protein output was low (*Figure 3D,F*). These trends were also seen when the other parameters were fixed at different values. Thus, the model predicted qualitatively different noise profiles in protein data. The noise profile most similar to the experimentally observed profile was the one in which $k_{on}$ varies between cells to generate a range of protein output. We surmise from these results that transcription of *sens* at the DV boundary of the wing disc might be regulated by modulating promoter burst frequency via $k_{on}$. Indeed, analysis of mRNA expression in the wing disc by single-molecule fluorescence in situ hybridization (smFISH) indicates that burst frequency is modulated for the *sens* gene (*Bakker et al., 2020*). Burst frequency modulation has also been observed for other developmental genes (*Bakker et al., 2020*; *Bartman et al., 2019*; *Fukaya et al., 2016*; *Zoller et al., 2018*).

Results from the mathematical model suggest that Sens protein noise comes from two distinct sources: (1) transcriptional bursting kinetics, and (2) mRNA and protein birth-death processes. The latter source generates a constant amplitude of protein fluctuations (Fano factor) across the entire spectrum of Sens output. When cells have a low transcription burst frequency, they experience larger fluctuations in mRNA numbers, which transmits to larger protein fluctuations, and generates the Fano factor peak. As promoter activation events become more frequent, they approximate a constitutively active promoter such that RNA-protein birth-death processes dominate the noise, and the Fano factor drops to a constant level.

When we implemented this more complex model to simulate the loss of miR-9a regulation, the result was similar to the experimental increase in Fano factor (*Figure 4D,E*). This result was observed whether the translation rate ($S_p$) or mRNA decay rate ($D_m$) parameter was altered 1.8-fold to simulate loss of miR-9a repression.

## Experimental validation of model predictions on transcription bursts and noise

The model predicts that transcription bursting is a significant contributor to Sens protein noise when cells contain fewer than 300 molecules (*Figure 3E*). To test this prediction, we landed the *sens* transgene in a different location of the genome. We reasoned that a different genomic neighborhood might change transcriptional bursting dynamics due to altered chromatin accessibility. We chose 57F5 to land *sens*, since both 22A3 and 57F5 are widely used landing sites for *Drosophila* transgenes (*Figure 5A*; *Venken et al., 2006*; *Venken et al., 2009*). There was no difference in the percentage of imaged cells positive for Sens between 22A3 (34.3 ± 3.7% positive) and 57F5 wing discs (34.7 ± 3.4% positive). The *sens* transgene inserted at 57F5 was also comparable to the 22A3 site in its ability to rescue the mutant endogenous *sens*, as well as express Sens protein in a similar pattern at the wing disc DV boundary (*Figure 5B–D*). Median Sens output was modestly higher (30–35%) when expressed from 57F5 (*Figure 5—figure supplement 1A,B*). However, the distribution of Sens output was not significantly different between 57F5 and 22A3 when normalized to the median value (*Figure 5—figure supplement 1D*). This result indicates that *sens* expression from 57F5 was uniformly higher across the entire spectrum of cells expressing the protein.

We generated animals where the alleles at 57F5 were singly tagged with sfGFP and mCherry (*Figure 5C,D*), and we quantitated the Fano factor after correction for measurement noise at 57F5. We then compared the Fano factor profile from the gene located at 57F5 versus 22A3 (*Figure 5E*). Strikingly, the Fano factor peak from the 57F5 gene was greatly increased in amplitude and width, and was maximal in cells with higher levels of Sens protein, compared to the peak from the 22A3 gene. However, the peak from 57F5 did relax to a constant Fano factor in cells with higher Sens output. This level was indistinguishable from the constant Fano level in 22A3 cells with comparable Sens output (*Figure 5E*). Thus, the 57F5 alleles do not appear to alter birth-death processes for *sens* mRNA and protein but do appear to affect some other process related to stochasticity.

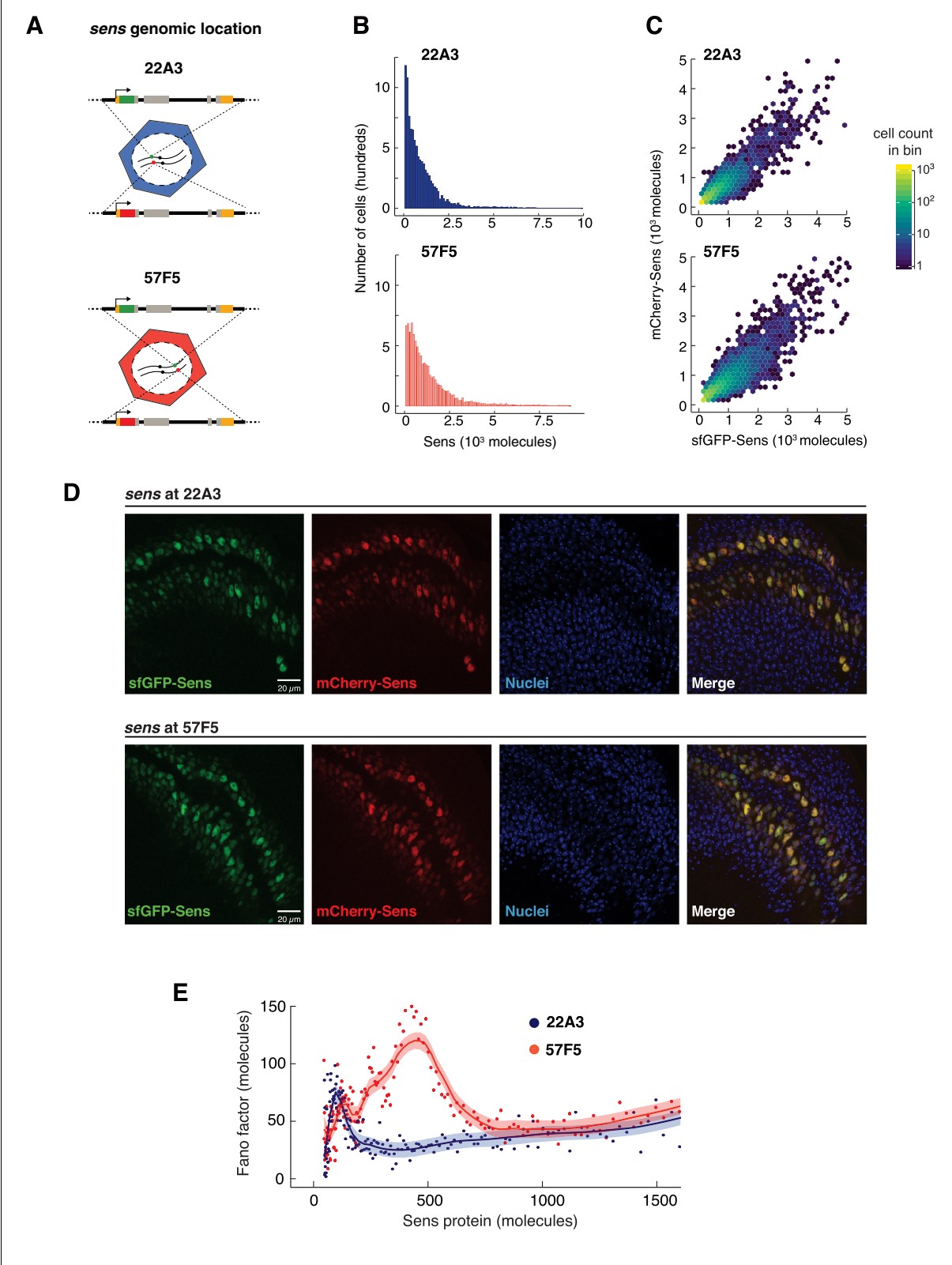

**Figure 5.** Genome location of the *sens* gene dramatically affects Sens noise. (A) The *sens* transgenes were inserted into one of two locations on chromosome II - 22A3 or 57F5. (B) Histograms depicting the frequency distribution of Sens protein number per cell. 10,000 cells were randomly sampled from 22A3 and 57F5 datasets each for comparison (see also *Figure 5—figure supplement 1*). (C) Scatter plot with hexagonal binning of single-cell sfGFP-Sens and mCherry-Sens protein numbers from *sens* genes inserted at 22A3 or 57F5. Randomly chosen subsets of 6,000 cells were

*Figure 5 continued on next page*

*Figure 5 continued*

plotted for each genotype for ease of comparison. Colors represent cell counts in respective hexagonal bins. (D) Confocal micrographic images of sfGFP- and mCherry-Sens fluorescence in wing discs expressing the *sens* gene inserted at 22A3 or 57F5. Nuclei are counterstained with DAPI (blue). Scale bars are 20 μm. Image brightness is enhanced (identically across 22A3 and 57F5) for ease of visualization. (E) The Fano factor profiles from cells expressing *sens* at 57F5 or 22A3. Cells with more than 800 molecules have identical Fano values at both genomic positions. The Fano peaks at lower Sens levels are very different between the genomic locations. Shaded regions are 95% confidence intervals.

The online version of this article includes the following figure supplement(s) for figure 5:

**Figure supplement 1.** Comparison of Sens expression profiles for various allelic pairs.

## Allele pairing at 57F5 generates *trans* regulation and enhanced noise

We looked for local properties of the genome at 22A3 and 57F5 that might be responsible for the different noise properties of *sens* inserted at those sites. Metazoan chromosomes are physically segregated into self-associating domains of chromatin called TADs (Topologically Associated Domains) (*Dixon et al., 2016*; *Szabo et al., 2018*). Domains vary in length, gene density and chromatin accessibility (*Dowen et al., 2014*). TADs are separated from each other by insulator sequences (*Ali et al., 2016*; *Stadler et al., 2017*; *Van Bortle et al., 2014*). Using published Hi-C and ChIP-seq data for the *Drosophila* genome (*Stadler et al., 2017*), we determined that the *sens* insertion site at 22A3 is in the middle of a large TAD, ~50 kb from its 5' and 3' insulators (*Figure 6—figure supplement 1A*). In contrast, the *sens* insertion site at 57F5 is in a small TAD, ~1 kb from its 3' insulator (*Figure 6—figure supplement 1B*).

Insulators physically associate with one another, leading to altered chromatin configurations (*Yang and Corces, 2011*). When insulators associate in cis, they form loops along a chromosome. DNA loops alter enhancer interactions with *cis* promoters or prevent the spread of heterochromatin into looped regions (*Fujioka et al., 2016*; *Yang and Corces, 2011*). In contrast, when insulators associate in trans, they facilitate enhancer regulation of promoters on other chromosomes by bringing them into close proximity (*Fujioka et al., 2016*; *Lim et al., 2018*; *Piwko et al., 2019*). Such *trans* regulatory phenomena, called transvection, appear to be a mode of gene regulation in several species including humans (*Hark et al., 2000*; *Liu et al., 2008*; *Masui et al., 2011*; *Rassoulzadegan et al., 2002*). In *Drosophila*, homologous chromosomes are extensively paired in somatic cells throughout most life stages (*Metz, 1916*), leading to physical co-localization of paired alleles in nuclei. This sometimes leads to *trans* regulation of one allele by its paired allele due to dynamic inter-TAD contacts in trans (*Szabo et al., 2018*). Since the *sens* gene was positioned either close to or distant from a TAD insulator in 57F5 or 22A3 respectively, we wondered if insertion altered the *cis* or *trans* regulation of *sens*.

To test this hypothesis, we examined protein output from the *mCherry-sens* allele alone. When this allele was placed at 22A3 in trans to a *sfGFP-sens* allele at 22A3, a unimodal distribution of mCherry-Sens protein was observed (*Figure 6A*). A strikingly different distribution was observed when *mCherry-sens* was placed at 57F5 in trans to a *sfGFP-sens* allele at 57F5 (*Figure 6A*). A sizable fraction of cells expressed higher levels of mCherry-Sens, creating a broader, bimodal distribution. We then placed a 57F5 *mCherry-sens* allele in trans to a 22A3 *sfGFP-sens* allele (*Figure 6A*). If enhanced expression of *mCherry-sens* at 57F5 was dependent on *trans* regulation between paired alleles, then unpaired 57F5/22A3 cells would behave like 22A3/22A3 cells. Conversely, if enhanced *mCherry-sens* expression was due to *cis* regulation at 57F5, then the mCherry-Sens output from 57F5/22A3 cells would behave like 57F5/57F5 cells. Strikingly, the observed distribution of mCherry-Sens from 57F5/22A3 cells was identical to that of 22A3/22A3 cells (*Figure 6A*). A similar result was observed when sfGFP-Sens output was monitored (data not shown). Single-molecule FISH analysis of *sens* nascent RNA confirmed that alleles positioned at the same locus (either 22A3 or 57F5) were physically co-localized in nuclei, whereas alleles positioned at heterologous loci were not co-localized (*Bakker et al., 2020*). Thus, even though *sens* alleles are physically paired at either 22A3 or 57F5, they regulate one another in trans when located at 57F5 but not at 22A3.

We then asked whether *cis* or *trans* regulation of *sens* at 57F5 is responsible for the enhanced noise in Sens protein output observed in 57F5/57F5 cells (*Figure 5E*). We generated animals with the *mCherry-sens* allele at 57F5 and the *sfGFP-sens* allele at 22A3, and quantitated the Fano factor after correction for measurement noise (*Figure 6B*). If *cis* regulation of *sens* at 57F5 was responsible

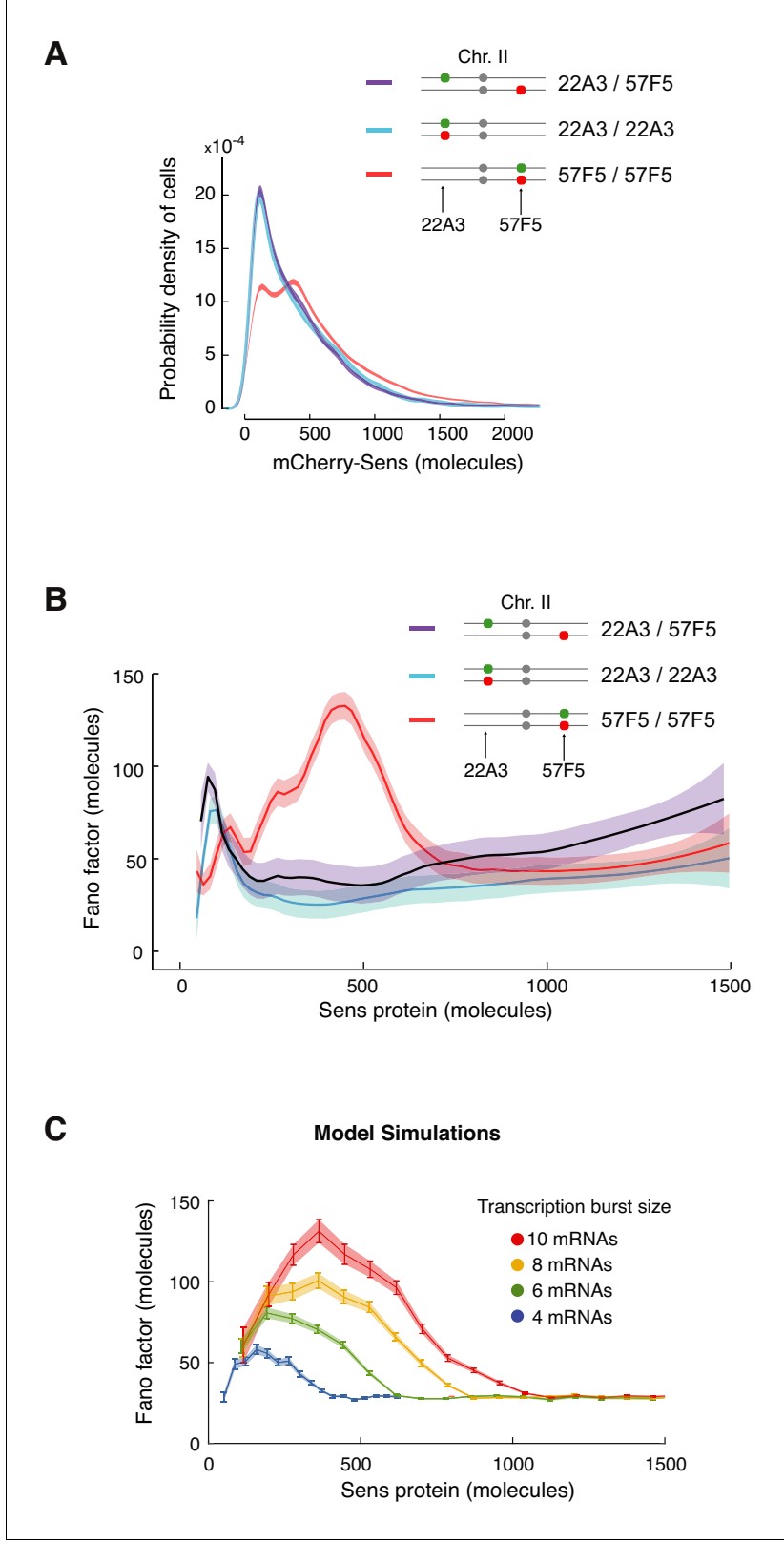

**Figure 6.** Enhanced protein noise requires *trans* regulation at 57F5. (**A**) Frequency distribution of mCherry-Sens protein number in cells. The *mCherry-sens* allele is located at either 22A3 or 57F5 as indicated. The *sfGFP-sens* allele is either paired or unpaired with the *mCherry-sens* allele as indicated. Lines are 95% confidence intervals of moving averages (see also **Figure 6—figure supplement 1**). (**B**) The Fano factor of Sens from cells expressing

*Figure 6 continued on next page*

*Figure 6 continued*
Sens protein either from paired alleles (22A3/22A3 and 57F5/57F5) or unpaired alleles (22A3/57F5). Moving line averages are shown, and shaded regions are 95% confidence intervals. (C) Model simulations in which transcription burst size (defined as $Sm/koff$) is set to different values as shown. The Fano peak amplitude and position change as burst size varies, but all relax to a constant basal level. These trends are similar to those observed in (B). Error bars and shaded regions are 95% confidence intervals (see also *Figure 6—figure supplement 2*).

The online version of this article includes the following figure supplement(s) for figure 6:

**Figure supplement 1.** Chromatin landscape of genomic loci 22A3 and 57F5 determined from Hi-C data.
**Figure supplement 2.** Fano factor profiles from model simulations in which *sens* alleles at 57F5 are regulated independently in cis or there is promoter inhibition in trans.

for the enhanced noise, then the Fano factor from 57F5/22A3 cells would still be high (*Figure 6— figure supplement 2A*). However, the observed Fano factor from 57F5/22A3 cells resembled that from 22A3/22A3 and not 57F5/57F5 cells (*Figure 6B*). This result strongly suggests that the enhanced noise is due to *trans* regulation of *sens* at 57F5.

We turned to our modeling framework to elucidate how *trans* regulation might enhance noise in protein output. Paired alleles sometimes inhibit each other's transcription output (*Lim et al., 2018*). Such inhibition would lead to greater differences in allelic output of protein, which might explain the enhanced Fano factor. We tested this by simulating *trans*-inhibitory alleles. When one allele's transcriptional state was 'on', the rate of activation ($k_{on}$) of the other allele was decreased. While this resulted in an increased amplitude in the Fano peak, it did not affect peak width and it did not shift the peak position to greater Sens output (*Figure 6—figure supplement 2B*). Both of these latter features were observed experimentally (*Figure 6B*). We then considered a simpler scenario in which *trans* regulation increased transcription, as suggested by the modest increase in protein output from paired 57F5 alleles (*Figure 6A*, *Figure 5—figure supplement 1A,B*). When the transcriptional parameters in the model were varied, we found that a small increase in transcription burst size could capture the effect of changing *sens* gene location from 22A3 to 57F5 on the Fano factor (*Figure 6C*). This suggests that changing burst size by *trans* regulation might account for the dramatic effects on Sens protein noise.

## Sensory bristle patterns become disordered by enhanced *sens* noise

Stripes of cells 4–5 cell diameters wide are induced to express Sens by Wg at the DV boundary (*Alexandre et al., 2014*; *Eivers et al., 2009*; *Phillips and Whittle, 1993*). Within each stripe, cells near the center undergo lateral inhibition, and consequently, some of these upregulate Sens to become S cells (*Alexandre et al., 2014*; *Troost et al., 2015*). This pattern was experimentally observed whether *sens* was transcribed from the 22A3 or 57F5 locus (*Figures 5D* and *7A*).

Many cells in the center of stripes contain 500–1000 Sens molecules (*Figure 7A* and *Figure 7— figure supplements 1A* and *2*). Thus, cells undergoing lateral inhibition are comparable in Sens protein number to those cells with highest noise from the 57F5 gene. To confirm that these central cells were high-noise for 57F5, we mapped the values of each cell's Fano factor to their spatial positions within wing discs. For the 57F5 gene, cells with abnormally high noise were located throughout the stripe, including the center from which S cells are chosen (*Figure 7B* and *Figure 7—figure supplement 1B*). Thus, it is highly likely that some 57F5 cells undergoing lateral inhibition were experiencing enhanced fluctuations in Sens protein number. In contrast, 22A3 cells with the highest noise were located at the edges of each stripe, distant from the central region from which S cells normally emerge (*Figure 7B* and *Figure 7—figure supplement 1B*). Thus, 22A3 cells experiencing the largest fluctuations contain very low Sens protein and are unlikely to adopt S-cell fates. We had observed that only paired alleles at 57F5 caused Sens noise enhancement, and unpaired alleles at 57F5 and 22A3 did not. When we mapped 57F5/22A3 cells with the highest noise, they were largely restricted to the edge of stripes closest to the DV boundary (*Figure 7—figure supplement 1B*).

We reasoned that if cells undergoing lateral inhibition experienced a large enough fluctuation in Sens number, then an error in cell fate determination might occur. We had measured Sens protein in cells undergoing decisions to make chemosensory bristles. Chemosensory bristles are periodically positioned in a row near the adult wing margin, such that approximately every fifth cell is a bristle

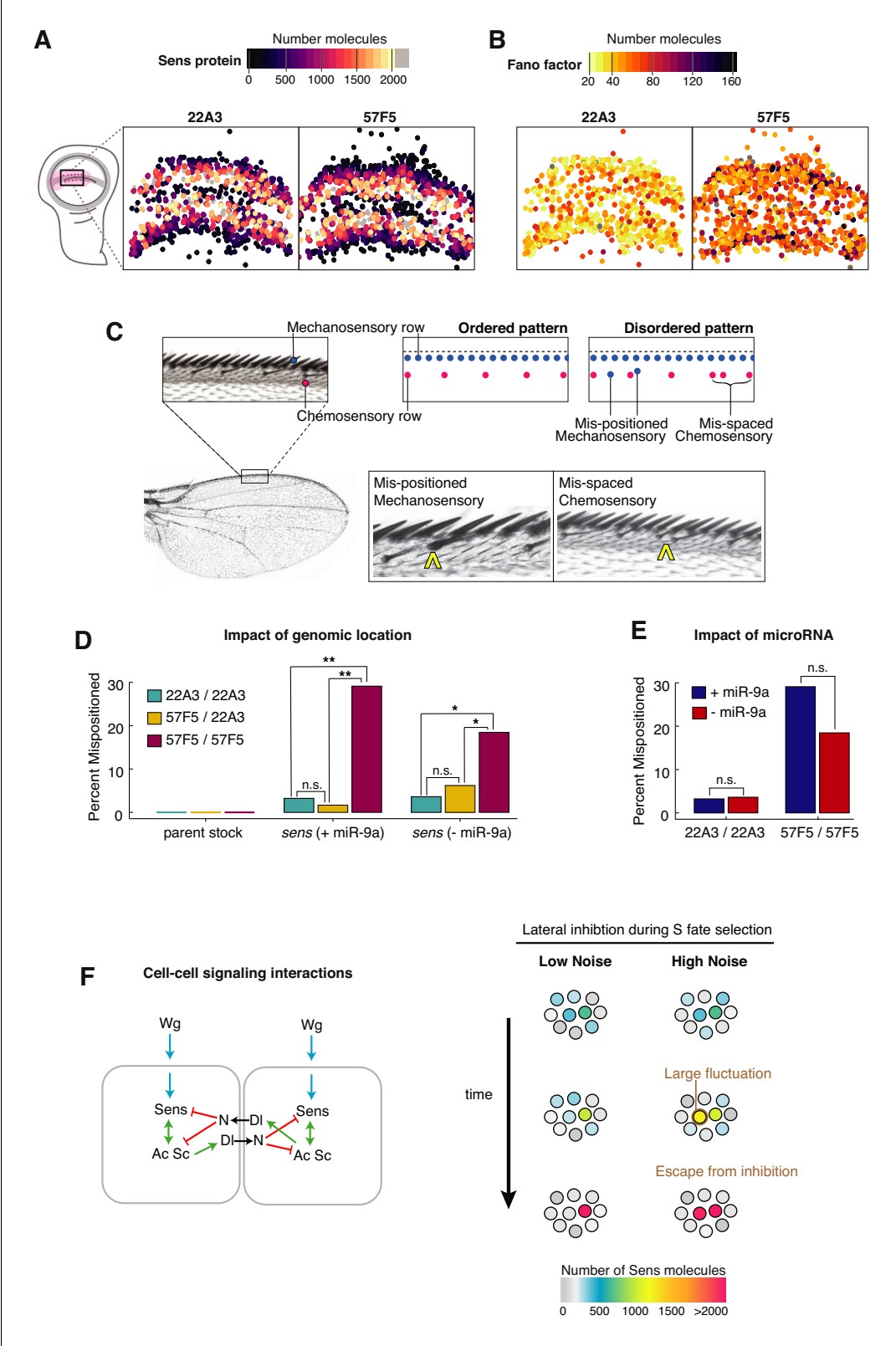

**Figure 7.** Self-organized sensory patterning is disrupted by stochastic gene expression. (A, B) The centroids of Sens-positive cells in 22A3/22A3 and 57F5/57F5 wing discs are mapped and color coded according to Sens protein number (A), and Fano factor level (B). Cells exhibiting high Fano factor values are distributed throughout the proneural zone of the 57F5/57F5 disc, where S fate determination occurs (see also *Figure 7—figure supplements 1* and *2*). (C) The dorsal surface of the adult wing margin displays two ordered rows of sensory organs - an outer continuous row of thick

*Figure 7 continued on next page*

*Figure 7 continued*

mechanosensory bristles (blue) and an inner periodic row of thin chemosensory bristles (magenta). Disorganized patterns are observed when bristles are incorrectly positioned. Instances of ectopic (mis-positioned) mechanosensory bristles in the chemosensory row (center) and ectopic chemosensory bristles, which disrupt periodic spacing (right), were observed and counted. (D) Percentage of adult wings with one or more patterning error. Pattern disorder is much greater when *sens* alleles are expressed from 57F5/57F5 relative to alleles at 22A3/22A3 or 57F5/22A3. This result is observed regardless of whether miR-9a regulates *sens* or not. Genotypes were compared by calculating the odds ratio of mispatterning, and determined to be significantly different from one if p<0.05 using a Fischer's exact test: n.s., not significant; *p<0.05; **p<.005 (*Figure 7—figure supplement 3*). (E) Uniformly increasing Sens protein number 1.8-fold by removing miR-9a regulation does not lead to greater pattern disorder. Genotypes were compared by calculating the odds ratio of mis-patterning, and determined to be significantly different from one if p<0.05 using a Fischer's exact test: n. s., not significant (*Figure 7—figure supplement 3*). (F) Left: A schematic outlining the induction of *sens* by Wg (blue) followed by *sens* auto-activation through feedback with Ac Sc (green) and through mutual inhibition between neighboring cells (red). Right: When expression noise is sufficiently small, Sens levels progressively increase or decrease in neighboring cells. However, when noise is enhanced, a large fluctuation will alter the trajectory of Sens expression towards an erroneous outcome.

The online version of this article includes the following source data and figure supplement(s) for figure 7:

**Source data 1.** Odds ratio of wings with mispositioned mechanosensory bristles.
**Figure supplement 1.** Maps of Sens protein number and Fano factor in replicate wing discs.
**Figure supplement 2.** Average Sens expression as a function of cell position relative to the DV boundary.
**Figure supplement 3.** Chemosensory bristle spacing is dependent on genomic location of the *sens* gene.

---

(*Figure 7C*). Mechanosensory bristles form in a continuous row most proximal to the adult wing margin, and they are selected 8–10 hr after the chemosensory cells are selected (*Hartenstein and Posakony, 1989*). Thus, mechanosensory bristles positioned incorrectly in the chemosensory row might be derived from proneural cells that escaped lateral inhibition during chemosensory specification (*Figure 7C*).

We quantified ectopic bristles in 57F5 versus 22A3 adults and determined the frequency with which a wing contained one or more ectopic bristles. The frequency was ten-fold higher in 57F5/57F5 adults compared to 22A3/22A3 adults (*Figure 7D*, *Figure 7—source data 1*). Indeed, an error was seen in one of three adult wings. Consistent with this, chemosensory bristles were also more frequently specified in 57F5 adult wings compared to 22A3 (*Figure 7C* and *Figure 7—figure supplement 3*). The increase in pattern disorder was not due to disruption of genes residing in the 57F5 TAD since none are annotated as neurogenic (*Cassidy et al., 2013*). Nor was it due to genetic background in the different lines since the 22A3 and 57F5 parental stocks had identical chemosensory and mechanosensory bristle frequencies (*Figure 7D*, *Figure 7—figure supplement 3* and *Figure 7—source data 1*). Moreover, adults with unpaired alleles (57F5/22A3) had an ectopic bristle frequency of 1.7%, not significantly different from 22A3/22A3 adults (*Figure 7D* and *Figure 7—source data 1*). Thus, pairing between alleles at 57F5 caused greater pattern disorder.

The pairing of *sens* at 57F5 led to a modest increase in the level of Sens output and a dramatic increase in the noise of Sens output. The greater pattern disorder could be due to the enhanced noise or level or both. To distinguish between these possibilities, we analyzed patterning of the adult wing margin when *sens* was rendered insensitive to miR-9a repression. Loss of miR-9a repression increased Sens protein levels 80–85% in all proneural cells (*Figure 4C* and *Figure 5—figure supplement 1A,C,E*) and modestly increased protein noise (*Figure 4D*). If pattern disorder was caused by enhanced protein levels, we reasoned that loss of miR-9a repression would generate even greater pattern disorder than genome position, since protein levels were only increased ~35% in 57F5/57F5 cells (*Figure 5—figure supplement 1A,B,D*). However, loss of miR-9a repression did not increase the frequency of patterning errors in adult wings (*Figure 7E*, *Figure 7—figure supplement 3*, and *Figure 7—source data 1*). In a different perspective, the miR-9a mutant *sens* paired at 22A3 expressed 36% more protein than the wildtype *sens* paired at 57F5 (*Figure 5—figure supplement 1A*). However, its phenotypic error frequency was 3.6% in contrast to 29.1% for wildtype *sens* at 57F5 (*Figure 7E*). The most parsimonious explanation is that bristle pattern disorder in 57F5/57F5 animals is caused by the greatly enhanced noise in *sens* gene expression.

## Discussion

An outstanding challenge is to determine whether stochasticity inherent to gene expression is transmitted across scales to vary the fidelity of pattern formation. We have focused on the organization of sensory bristles along the adult wing margin. During pattern formation, cells experience noise in Sens protein copy number that derives from two sources. One source is from the discontinuous bursts of *sens* transcription, and the other source is from random birth-death events that affect *sens* mRNA and protein. For *sens*, as defined by the 19.2 kb region constituting the transgene, this intrinsic noise was not sufficient to transmit disorder to the adult pattern. However, when *sens* was subject to *trans* regulation between paired alleles, protein noise was greatly enhanced, which was sufficient to disorder the adult pattern. Therefore, *trans* interaction between paired homologs is an unanticipated source of noise. It is tempting to speculate that *trans* regulation might be a natural means to modulate gene expression noise.

Allelic pairing has been observed across multiple organisms (*Hark et al., 2000*; *Liu et al., 2008*; *Rassoulzadegan et al., 2002*). Pairing often precedes *trans* regulatory interactions, such as X-inactivation (*Masui et al., 2011*). In *Drosophila*, pairing and *trans* regulation between homologs has been demonstrated for several developmental genes (*Duncan, 2002*; *Fukaya and Levine, 2017*; *Johnston and Desplan, 2014*; *Lunde et al., 1998*). Indeed, *trans* allelic interactions appear to be a pervasive feature of the *Drosophila* genome (*Bateman et al., 2012*; *Blick et al., 2016*; *Mellert and Truman, 2012*). However, pairing of homologous alleles is not necessarily indicative of *trans* regulation (*Viets et al., 2019*).

It remains to be determined how pairing at 57F5 enhances Sens output and noise. The noise profile observed for paired *sens* alleles at 57F5 can be partly modeled as the effect of enhanced transcription burst size (*Figure 6*). Since the two-state model uses general rate parameters $k_{on}$, $k_{off}$, and $S_m$ to regulate bursting, it is agnostic to the specific molecular processes that direct transcription. Distinct mechanisms such as chromatin remodeling, enhancer looping, transcription factor binding-unbinding, or preinitiation complex assembly-disassembly might be rate-limiting for $k_{on}$, $k_{off}$, or $S_m$ at different levels of Sens expression. Thus, the complex noise profile observed for 57F5 might be a signature of multi-state transcription. Indeed, multi-state transcription kinetics have been observed for other genes (*Bothma et al., 2014*; *Rodriguez et al., 2019*; *Tantale et al., 2016*). It is also possible that certain types of mRNA state transitions could generate an enhanced profile of protein noise. These might include transitions from an unspliced to spliced state (*Wan and Larson, 2018*), cytoplasmic mRNA processing (*Hansen et al., 2018*), differential translation efficiencies, or toggling between reversible translating and non-translating states (*Yan et al., 2016*). Therefore, although our modeling implicates transcriptional bursting as a major source of protein noise, it remains to be experimentally verified. Our use of transgenes inserted into only two sites should be augmented with comparison to more sites and to the endogenous *sens* gene. Examining transcription bursting via smFISH or MS2-MCP tagged RNA at paired and unpaired loci will be necessary to determine if bursting kinetics are different for different genomic locations.

Noise levels differ between 22A3 and 57F5 cells only in the expression regime of 300–800 Sens molecules. Yet, this difference appears to affect pattern order-disorder. It would suggest that a subset of cells with fewer than 800 Sens molecules are at a developmental decision point between E and S fates. This is an order of magnitude less than the approximately 8,000–10,000 Sens molecules observed in terminal S-fated cells. Stochastic fluctuations have the largest impact when protein copy numbers are low. Yet, production of a large number of proteins would raise the time and metabolic cost required to undergo developmental transitions (*Rodenfels et al., 2019*; *Wagner, 2005*). It suggests that expression noise of certain genes is optimized to allow accurate pattern formation without requiring the production of large numbers of fate-determining proteins.

How might this optimization be realized? Individually varying the different rate constants in our two-state model produces very different protein noise-output relationships (*Figure 3*). Regulating $k_{on}$ provides the most effective way to increase protein output without increasing noise. Increasing $k_{on}$ increases the frequency of transcriptional bursts without increasing burst size. Indeed, smFISH experiments show that while *sfGFP-sens* transcription burst size is constant across the wing margin, mRNA output is regulated by tuning $k_{on}$ and burst frequency across cells (*Bakker et al., 2020*). We have inferred a similar regulatory mechanism for *sens* transcription by coupling protein noise measurements to stochastic models.

Proper inference of transcription kinetics using protein measurements requires a straightforward correlation between mRNA and protein numbers. Such correlations have been noted (*Raj et al., 2006*). For instance, *bicoid* mRNA and protein numbers are reproducible to within 10% and scale proportionately with gene dosage (*Gregor et al., 2007*; *Petkova et al., 2014*). Stochastic models of transcription and protein production were used to correctly infer mRNA and protein copy numbers for bacteriophage lambda repressor CI (*Sepúlveda et al., 2016*) and the HIV-1 Long Terminal Repeat promoter (*Dar et al., 2016*; *Dey et al., 2015*). Indeed, protein reporters have been successfully used to infer transcriptional bursting parameters $k_{on}$ and $k_{off}$ for a wide variety of transgenic and endogenous mammalian genes (*Suter et al., 2011*).

Our results suggest that pattern disorder is driven by Sens protein noise rather than protein levels. This might seem counterintuitive since there are many examples of gene overexpression causing developmental phenotypes. The explanation likely lies in the mechanism of wing margin patterning, which occurs in two stages (*Figure 7F*). First, the Wg morphogen induces Sens expression leading to tens to hundreds of protein molecules per cell (*Jafar-Nejad et al., 2006*). Second, Sens expression is self-organized into a periodic row of S and E cells by Delta-Notch mediated lateral inhibition (*Hartenstein and Posakony, 1990*; *Heitzler and Simpson, 1991*). If all cells should express Sens to an abnormally high level, lateral inhibition still acts on the relative differences between cells to properly resolve the pattern (*Corson et al., 2017*). Consistent with this, reducing endogenous *sens* gene dose to one copy does not affect bristle pattern formation (*Jafar-Nejad et al., 2006*). On the other hand, enhanced fluctuations in Sens output appear to cause pattern disorder. The simplest interpretation is that during the second stage, cell-to-cell transmission and reception of inhibitory signals is distorted by high intrinsic fluctuations in Sens (*Figure 7F*). If fluctuations are large enough to trigger positive feedback between proneural factors, it would render a cell resistant to lateral inhibition. The net outcome would be cells that spontaneously adopt S fates out of order.

Stochastic transcriptional fluctuations have been harnessed in bet hedging systems such as induction of lactose metabolism in *E. coli* (*Choi et al., 2008*) and cell competence in *B. subtilis* (*Süel et al., 2007*). In mammalian cells, stochastic fluctuations confer phenotypes such as HIV latency periods (*Weinberger et al., 2005*) and acquisition of cancer drug resistance (*Shaffer et al., 2017*). In developmental contexts, variability in transcription factor output has been shown to affect cell-fate switches that rely on absolute concentration thresholds (*Gregor et al., 2007*; *Raj et al., 2010*). However, some developmental systems buffer against expression stochasticity by relying on relative changes in expression or intercellular signaling (*Sonnen and Aulehla, 2014*). We show that a patterning system relying on lateral inhibition can buffer against tissue-scale changes in protein output, but is sensitive to stochastic fluctuations in protein copy numbers.

# Materials and methods

## Key resources table

| Reagent type (species) or resource | Designation | Source or reference | Identifiers | Additional information |
|---|---|---|---|---|
| Gene (*Drosophila melanogaster*) | *white*[1118] | Bloomington *Drosophila* Stock Center | *BDSC*: 3605 *Flybase*: FBst0003605 | |
| Gene (*Drosophila melanogaster*) | *sens*[E1] | *Nolo et al. (2001)* | *Flybase*: FBal0098024 | From Hugo Bellen |
| Gene (*Drosophila melanogaster*) | *sens*[E2] | Bloomington *Drosophila* Stock Center | *BDSC*: 5311 *Flybase*: FBal0098023 | |
| Strain, strain background (*Drosophila melanogaster*) | *y*[1] *w*[1118]; PBac{*y*[+]-attP-3B} VK00037 | Bloomington *Drosophila* Stock Center | *BDSC*: 9752 *Flybase*: FBst0009752 | 22A3 |

*Continued on next page*

*Continued*

| Reagent type (species) or resource | Designation | Source or reference | Identifiers | Additional information |
|---|---|---|---|---|
| Strain, strain background (*Drosophila melanogaster*) | *y¹ w¹¹¹⁸;* PBac{*y⁺*-attP-9A} VK00022 | Bloomington *Drosophila* Stock Center | *BDSC*: 9740 *Flybase*: FBst0009740 | 57F5 |
| Genetic reagent (*Drosophila melanogaster*) | Wildtype *sfGFP-sens [22A3]* | *Venken et al. (2006)*. From Hugo Bellen | | Pacman construct containing *sens* gene with N-terminal 3xFlag-TEV-StrepII-sfGFP-FlAsH fusion tag inserted at 22A3 |
| Genetic reagent (*Drosophila melanogaster*) | Mutant *sfGFP-sens [22A3]* | This paper | | *sens* transgene with N-terminal 3xFlag-TEV-StrepII-sfGFP-FlAsH fusion tag and two miR-9a binding sites mutated inserted at 22A3 |
| Genetic reagent (*Drosophila melanogaster*) | Wildtype *mCherry-sens [22A3]* | This paper | | *sens* transgene with N-terminal 3xFlag-TEV-StrepII-mCherry-FlAsH tag inserted at 22A3 |
| Genetic reagent (*Drosophila melanogaster*) | Mutant *mCherry-sens [22A3]* | This paper | | *sens* transgene with N-terminal 3xFlag-TEV-StrepII-mCherry-FlAsH fusion tag and two miR-9a binding sites mutated inserted at 22A3 |
| Genetic reagent (*Drosophila melanogaster*) | Tandem tag *sfGFP-mCherry -sens [22A3]* | This paper | | *sens* transgene with N-terminal 3xFlag-TEV-StrepII-mCherry-sfGFP -FlAsH fusion tag inserted at 22A3 |
| Genetic reagent (*Drosophila melanogaster*) | Wildtype *sfGFP-sens [57F5]* | This paper | | *sens* transgene with N-terminal 3xFlag-TEV-StrepII-sfGFP-FlAsH fusion tag inserted at 57F5 |
| Genetic reagent (*Drosophila melanogaster*) | Mutant *sfGFP-sens [57F5]* | This paper | | *sens* transgene with N-terminal 3xFlag-TEV-StrepII-sfGFP-FlAsH fusion tag and two miR-9a binding sites mutated inserted at 57F5 |
| Genetic reagent (*Drosophila melanogaster*) | Wildtype *mCherry-sens [57F5]* | This paper | | *sens* transgene with N-terminal 3xFlag-TEV-StrepII-mCherry-FlAsH tag inserted at 57F5 |
| Genetic reagent (*Drosophila melanogaster*) | Mutant *mCherry -sens [57F5]* | This paper | | *sens* transgene with N-terminal 3xFlag-TEV-StrepII-mCherry-FlAsH fusion tag and two miR-9a binding sites mutated inserted at 57F5 |
| Genetic reagent (*Drosophila melanogaster*) | Tandem tag *sfGFP-mCherry - sens [57F5]* | This paper | | *sens* transgene with N-terminal 3xFlag-TEV-StrepII-mCherry-sfGFP -FlAsH fusion tag inserted at 57F5 |
| Antibody | Guinea Pig polyclonal anti-Sens | *Nolo et al. (2000)*. From Hugo Bellen | | IF(1:1000) |
| Antibody | Goat anti-guinea pig IgG Alexa 488 | Invitrogen | Cat# A-11073, RRID: AB_2534117 | IF(1:250) |
| Antibody | Mouse monoclonal anti-Flag, clone M2 | Sigma | Cat# F1804, RRID: AB_262044 | Elisa (1:100) |
| Antibody | Rabbit polyclonal anti-GFP | Molecular Probes | Cat# A-11122, RRID: AB_221569 | Elisa (1:5000) |
| Antibody | Goat polyclonal anti-rabbit IgG – HRP conjugated | GE Healthcare | Cat# RPN4301, RRID: AB_2650489 | Elisa (1:5000) |
| Recombinant DNA reagent | P[acman] wild type *sfGFP-sens* | *Venken et al. (2006)*. Gift of Koen Venken | | |

*Continued on next page*

*Continued*

| Reagent type (species) or resource | Designation | Source or reference | Identifiers | Additional information |
|---|---|---|---|---|
| Recombinant DNA reagent | P[acman] mutant *sfGFP-sens* | This paper | | 19.2 kb *D. melanogaster* genomic fragment including *sens* gene with miR-9a binding sites mutated and N-terminal fusion to sfGFP |
| Recombinant DNA reagent | P[acman] wild type *mCherry-sens* | This paper | | 19.2 kb *D. melanogaster* genomic fragment including *sens* gene with N-terminal fusion to mCherry |
| Recombinant DNA reagent | P[acman] miR-9a binding site mutant *mCherry-sens* | This paper | | 19.2 kb *D. melanogaster* genomic fragment including *sens* gene with miR-9a binding sites mutated and N-terminal fusion to mCherry |
| Recombinant DNA reagent | P[acman] wild type tandem tag *sfGFP-mCherry-sens* | This paper | | 19.2 kb *D. melanogaster* genomic fragment including *sens* gene with miR-9a binding sites mutated and N-terminal fusion to mCherry and sfGFP |
| Sequence-based reagent | 18S - Forward | This paper | PCR primers | CTGAGAAACGGCTACCACATC |
| Sequence-based reagent | 18S - Reverse | This paper | PCR primers | ACCAGACTTGCCCTCCAAT |
| Sequence-based reagent | Rpl21 - Forward | This paper | PCR primers | CTTGAAGAACCGATTGCTCT |
| Sequence-based reagent | Rpl21 - Reverse | This paper | PCR primers | CGTACAATTTCCGAGCAGTA |
| Sequence-based reagent | Sens - Forward | This paper | PCR primers | CAGGAATTTCCAGTGCAAACAG |
| Sequence-based reagent | Sens - Reverse | This paper | PCR primers | CGCCGGTATGTATGTACGTG |
| Sequence-based reagent | Hsp70Ba - Forward | This paper | PCR primers | AGTTCGACCACAAGATGGAG |
| Sequence-based reagent | Hsp70Ba - Reverse | This paper | PCR primers | GACTGTGGGTCCAGAGTAGC |
| Commercial assay or kit | 1-Step Ultra TMB-ELISA | Thermo Fisher | Cat #34028 | For Elisa assays |
| Chemical compound, drug | Paraformaldehyde (powder) | Polysciences | 00380–1 | |
| Chemical compound, drug | Triton X-100 | Sigma Aldrich | T9284-500ML | |
| Chemical compound, drug | VectaShield | Vector Labs | H-1000 | |
| Chemical compound, drug | 4',6-diamidino-2-phenylindole (DAPI) | Life Technologies | D1306 | |
| Software, algorithm | MATLAB script to automatically segment disc nuclei | *Peláez et al. (2015)* | | https://github.com/ritika-giri/stochastic-noise |
| Software, algorithm | MATLAB scripts for modeling simulations | This paper | | https://github.com/ritika-giri/stochastic-noise/tree/master/MATLAB%20scripts%20for%20gene%20expression%20simulation |
| Software, algorithm | R scripts for imaging data analysis | This paper | | https://github.com/ritika-giri/stochastic-noise/tree/master/R%20script%20for%20data%20analysis |
| Other | WM1 medium | *Restrepo et al. (2016)* | | Growth medium for organ culture |

## Experimental model and subject details

For all experiments, *Drosophila melanogaster* was raised using standard lab conditions and food. Stocks were either obtained from the Bloomington Stock Center, from listed labs, or were derived in our laboratory (RWC). A list of all mutants and transgenics used in this study is in the Key Resources Table. All experiments used female animals unless stated otherwise. The sample sizes were not computed when the study was designed. Sample sizes were determined such that >12,000 cells were measured for each genotype, as detailed in the section Quantitation and Statistics. Since only ~1,000 cells could be segmented per disc sample,>12 discs were analyzed for each genotype.

N-terminal 3xFlag-TEV-StrepII-sfGFP-FlAsH tagged *sens,* originally generated from the CH322-01N16 BAC, was a kind gift from K Venken and H Bellen (*Venken et al., 2006*; *Venken et al., 2009*). It has been shown to rescue *sens*[E1] and *sens*[E2] mutations (*Cassidy et al., 2013*; *Venken et al., 2006*). To generate mCherry tagged *sens*, the sfGFP coding sequence in 3xFlag-TEV-StrepII-sfGFP-FlAsh was swapped out for mCherry by RpsL-Neo counter-selection (GeneBridges). The sfGFP-mCherry tandem tagged *sens* transgene was generated similarly by overlap PCR such that sfGFP and mCherry sequences were separated by a 12 amino acid $(GGS)_4$ linker. The *miR-9a* binding site mutant alleles of the tagged *sens* transgenes were created by deletion of the two identified binding sites in the 607 nt *sens* 3'' UTR as had been described previously (*Cassidy et al., 2013*) to generate *sens*[m1m2] mutant transgenes. Cloning details are available on request. All BACs were integrated at PBacy+-attP-3BVK00037 (22A3) and PBacy+-attP-9AVK00022 (57F5) landing sites by phiC31 recombination (*Venken et al., 2006*).Transgenic lines were crossed with *sens* mutant lines to construct stocks in which *sens* transgenes were present in a *sens*[E1]/*sensE*[E2] trans-heterozygous mutant background. Both mutant alleles show greatly reduced levels of anti-Sens staining in cells, and the *E2* allele is a nonsense mutant that cuts off the protein's DNA-binding domain.

## Method details

### Adult wing imaging and pattern analysis

Adult females from uncrowded vials were collected on eclosion and aged for 1–2 days before being preserved in 70% ethanol. Wings from preserved animals were plucked out with forceps and kept ventral side up on a glass slide. Approximately, 10 pairs of wings were arranged per slide using a thin film of ethanol to lay them flat. Left and right wings from the same animal were positioned next to each other. Once specimens were arranged as desired, excess ethanol was wiped away. A second glass slide was coated with heptane glue (10 cm$^2$ double sided embryo tape dissolved overnight in 4 ml heptane) and pressed down onto the specimen slide to affix them dorsal side up. Then wings were mounted in 70% glycerol in PBS and sealed with nail polish for imaging. Wings were imaged using an Olympus BX53 upright microscope with a 10x UPlanFL N objective in brightfield. To achieve optimal resolution, 8–10 overlapping images were taken for each wing and stitched together in Adobe Photoshop.

Wings with at least one mechanosensory bristle placed ectopically in or adjacent to the chemosensory bristle row were counted as mis-patterned. The proportion of mis-patterned wings was calculated for each genotype (n $\geq$ 60). Genotypes were compared by calculating the odds ratio of mispatterning and determined to be significantly different from 1 if p < 0.05 using Fischer's exact test. For chemosensory bristle density, wing images were used to identify and mark chemosensory bristles along the margin in Fiji. The Euclidean distance between successive bristles was measured and bristle density was calculated as the inverse of mean spacing. 95% confidence intervals were calculated by bootstrapping and bristle distributions across genotypes were compared statistically using a student's t-test.

### Fluorescence microscopy

All fluorescence microscopy experiments used white pre-pupal animals. The white pre-pupal stage was chosen because it is a major transition in the life cycle and lasts for only 45-60 minutes (*Bainbridge and Bownes, 1981*), ensuring a high degree of developmental synchronization. Further, wing margin chemosensory precursor selection was observed to be tightly linked to the transition from late third larval instar to pre-pupal stage. Wing discs from staged animals were dissected out in ice-cold Phosphate Buffered Saline (PBS). Discs were fixed in 4% paraformaldehyde in PBS for 20

minutes at 25C and washed with PBS containing 0.3% Tween-20 (PBS-Tween). Then they were stained with 0.5 µg/ml DAPI and mounted in Vectashield. Discs were mounted apical side up and imaged with identical settings using a Leica TCS SP5 confocal microscope. All images were acquired at 100x magnification at 2048 x 2048 resolution with a 75 nm x-y pixel size and 0.42 µm z separation. Scans were collected bidirectionally at 400 MHz and 6x line averaged in the red and green channels to detect mCherry and GFP, respectively. Wing discs of different genotypes were mounted on the same microscope slide and imaged in the same session for consistency in data quality.

For immunofluorescence, discs were dissected and fixed before incubating with the primary guinea pig anti-Sens antibody (gift from H. Bellen) diluted 1:1000 in PBS-Tween. Tissues were washed three times for 5–10 min each in PBS-Tween, and incubated with goat anti-guinea pig Alexa488 (diluted 1:250, Invitrogen) for 1 hr. After three washes in PBS-Tween, they were stained with DAPI, and mounted in VectaShield (Vector Labs) for imaging.

## Image quantification and analysis

### Cell segmentation

For each wing disc, five optical slices containing proneural cells were chosen for imaging and analysis. A previously documented custom MATLAB script was used to segment nuclei in each slice of the DAPI channel (*Peláez et al., 2015*; *Qi et al., 2013*). Briefly, high intensity nucleolar spots were smoothed out to merge with the nuclear area to prevent spurious segmentation. Next, cell nuclei were identified by thresholding based on DAPI channel intensity. Segmentation parameters were optimized to obtain nuclei with at least 100 pixels and no more than 4000 pixels. To estimate the accuracy of the automated segmentation procedure, we compared its results to a manually curated dataset of over 500 nuclei from randomly chosen optical slices. Approximately 95.1% of nuclei were correctly identified using this algorithm, with a false positive rate of 2.6% and false negative rate of 2.3%. For each nuclear area so identified, the average signal intensity for the sfGFP and mCherry channels was recorded along with the relative position of its centroid in x and y. Since segmentation was based exclusively on the nuclear signal, it identified all cells present in the imaged area (*Figure 1—figure supplement 2A*).

### Background fluorescence normalization

The majority of cells imaged did not fall within the proneural region and therefore displayed background levels of fluorescence scattered around some mean level (*Figure 1—figure supplement 2B*). Sens expressing cells were present in the right-hand tail of the distribution. The background was channel specific and varied slightly from disc to disc (*Figure 1—figure supplement 2C*). Therefore, we calculated the 'mean channel background' for each channel in each disc individually. We did this by fitting a Gaussian distribution to the population and finding the mean of that fit. In order to separate Sens positive cells, we chose a cut-off percentile based on the normal distribution, below which cells were deemed Sens negative. We set this cut-off at the 84th percentile for all analysis (*Figure 1—figure supplement 2D*).

This was determined empirically by mapping cell positions relative to the proneural region. At and above the 84th percentile, mapped cells followed the proneural striped pattern. Lowering the cut-off led to addition of cells randomly scattered across the imaging field. Increasing the cut-off led to progressive narrowing of the proneural stripes. From this we inferred the fluorescence level at 84th percentile as a tolerant but specific threshold to identify Sens positive cells. Thus, to normalize measurements across tissues and experiments, this value was subtracted from the total measured fluorescence for all cells in that disc and channel. Only cells with values above the threshold for both mCherry and sfGFP fluorescence were assumed Sens positive (usually 30% of total cells) and carried forward for further analysis (*Figure 1—figure supplement 2E*).

### mCherry and sfGFP fluorescence scaling

We required the relative fluorescence of the mCherry and sfGFP channels to be scaled in equivalent units. To do this, we fit a linear equation as shown, and derived best-fit values for slope and constant intercept.

$$RFU_{sfGFP} = Slope\left(RFU_{mCherry}\right) + Constant$$

To preserve data integrity, the slope and constant was calculated for each wing disc separately. Linear correlation coefficients were consistently high between mCherry and sfGFP fluorescence, ranging from 0.85 to 0.95. Finally, to rescale single cell mCherry fluorescence in units of sfGFP-Sens fluorescence, we applied the following transformation to each cell's raw mCherry intensity (*Figure 2—figure supplement 2A*).

$$ScaledRFU_{mCherry} = Slope(RFU_{mCherry}) + Constant$$

Once the two-channel RFUs were made equivalent, they were summed to obtain total Sens RFU for each cell as shown.

$$RFU_{Sens} = RFU_{sfGFP} + ScaledRFU_{mCherry}$$

## Fluorescent tag similarity

As an additional control, we checked by various means if indeed sfGFP-Sens and mCherry-Sens proteins behaved similarly in vivo such that the nature of the protein tag did not affect quantitative assays.

First, we measured the molecule counts of sfGFP-Sens and mCherry-Sens in the same cells using Fluorescence Correlation Spectroscopy (FCS). As can be seen in *Figure 1G*, we obtained similar numbers of Sens molecules irrespective of which fluorescent tag was attached. This indicated that both alleles express equal numbers of proteins in vivo.

Second, using the microRNA repression assay detailed in *Figure 4A–C*, we sensitively assayed whether the nature of the tag affects protein output quantitatively. If one tag were differentially expressed relative to the other, we would expect the fold-repression values calculated using mCherry tagged *sens* alleles to be different from sfGFP tagged *sens* alleles. This was not observed (*Figure 4—figure supplement 1*).

Third, to ensure that we did not under-estimate stochastic noise due to Fluorescence Resonance Energy Transfer (FRET), we imaged tandem tagged sfGFP-mCherry-Sens samples in both channels after exciting only the donor (sfGFP) molecules. There was negligible FRET from sfGFP to mCherry when using imaging parameters identical to experimental runs (*Figure 2—figure supplement 1*).

## Intrinsic noise and Fano factor calculation according to protein level

We used the following formula to calculate intrinsic noise (*Elowitz et al., 2002*). Mathematically, it is the variance remaining after the co-variance term of two variables is subtracted from their total variance. This value is then normalized to the squared mean ($\eta^2 = \sigma^2/\mu^2$) to obtain the following dimensionless quantity:

$$\eta^2_{intrinsic} = \frac{\left\langle (x-y)^2 \right\rangle}{2\langle x \rangle \langle y \rangle}$$

Here $x$ and $y$ represent number of sfGFP molecules and mCherry molecules, respectively, in a given cell. Angled brackets denote averages over a population of cells. This term provides a single value of intrinsic noise for the entire cell population. Since Sens expression varies over three orders of magnitude, we partitioned cells into small bins according to their Sens protein number ($x + y$). Sens RFU was log-transformed and we used a bin width of 0.02 log(RFU) to partition cells (*Figure 2—figure supplement 2B, C*). We then calculated intrinsic noise and mean Sens protein number for each binned sub-population. These were multiplied together to calculate the Fano factor for each bin.

$$Fano\,factor = \left(\frac{\sigma^2}{\mu}\right) = \eta^2 . \mu$$

Given that the number of cells in each bin was not constant, and that variance estimates are affected by sample size, we calculated confidence intervals around the calculated Fano factor for each bin by bootstrapping. We resampled bin populations 50,000 times with replacement. The 2.5th and 97.5th percentile estimates were used to construct a 95% confidence interval for that bin's Fano factor (*Figure 2—figure supplement 2D*).

## Measurement noise correction

Intrinsic noise and the Fano factor were calculated as described above for tandem-tagged *sfGFP-mCherry-sens* wing discs. The Fano factor profile was identical for tandem-tagged *sens* genes inserted at either 22A3 or 57F5. Therefore, we pooled data generated from both locations before binning into sub-populations. We expect the tandem-tag construct, along with our analysis pipeline, to account for lack of correlation in red-green fluorescence due to non-linearities in imaging and detection, differences between the two tags such as folding time and spectral properties, as well as increased variance due to image analysis or segmentation errors.

In order to construct a statistical model for measurement noise at each level of Sens output, we used a Lowess regression to fit a continuous line through the data (as seen in *Figure 2C*). The Lowess algorithm fits a locally weighted polynomial onto x-y scatter data and therefore does not rely upon specific assumptions about the data itself. The local window used to calculate a fit was kept constant for all Lowess fits. Using our statistical model, we generated a predicted Fano factor that was due to measurement noise for each bin. This predicted value was subtracted from the Fano factor that was due to both measurement and gene expression noise for each bin. The difference obtained is an estimate of the Fano factor due to noise in *sens* gene expression.

## FCS sample calibration and measurements

White pre-pupal wing discs were dissected in PBS and sunken into LabTek 8-well chambered slides containing 400 µl PBS per well (*Papadopoulos et al., 2019*). Discs were positioned such that the pouch region was facing the bottom of the well to be imaged. FCS measurements were made using an inverted Zeiss LSM780, Confocor 3 instrument with APD detectors. A water immersion 40x objective with numerical aperture of 1.2 (which is optimal for FCS measurements) was used throughout. Fast image scanning was utilized for identification of cell nuclei to be measured by FCS. Prior to each session, we used 10 nM dilute solutions of Alexa488 and CF586 dyes to calculate the average number of particles, the diffusion time and define the structural parameters $w_{xy}^2$ and $z_0$. Using these we calibrated the Observation Volume Element (OVE) whose volume can approximated by a prolate ellipsoid $\left(V_{OVE} = \pi^{\frac{3}{2}} w_{xy}^2 z_0\right)$. Measurements were performed in Sensory Organ Precursor cells (SOPs or S-fated), as well as first and second order neighbors, residing dorsally or ventrally of the S-fated cell (*Figure 1G*). Measurements were subjected to analysis and fitting, using a two components model for three-dimensional diffusion and triplet correction as follows:

$$G(\tau) = 1 + \frac{1}{N}\left(\frac{1-y}{\left(1+\frac{\tau}{\tau_{D_1}}\right)\sqrt{1+\frac{w_{xy}^2\tau}{w_z^2\tau_{D_1}}}} + \frac{y}{\left(1+\frac{\tau}{\tau_{D_2}}\right)\sqrt{1+\frac{w_{xy}^2\tau}{w_z^2\tau_{D_2}}}}\right)\left(1+\frac{T}{1-T}e^{-\frac{\tau}{\tau_T}}\right)$$

FCS measurements were excluded from analysis if they exhibited marked photobleaching or low CPM that is counts per molecule (CPM < 0.5 kHz per molecule per second). Due to the higher CPM of sfGFP, it was expected that Sens-sfGFP measurements are more accurate. We, nevertheless, observed fairly similar molecular numbers for both sfGFP-Sens and mCherry-Sens. Normalized auto correlation curves allowed us to compare the differential mobilities of the tagged Sens protein molecules in the nucleus and their degree of interaction with chromatin. Consistently, for both sfGFP and mCherry tagged transcription factors, we observed similar amplitudes and decay times of the slow FCS component, suggesting that the interaction with chromatin is not substantially different for differently tagged Sens molecules or even at different Sens concentrations.

We compared Sens protein concentrations as measured by FCS to single cell fluorescence data from confocal imaging of the fixed tissue (*Figure 1—figure supplement 3*). All comparisons were done for the genotype shown below since all FCS measurements were made in trans-heterozygous *mCherry-sens[22A3]*/*sfGFP-sens[22A3]* animals in a *sens* mutant (*sens^E1^/sens^E2^*) background.

## miR-9a repression measurements

In order to measure the fold-decrease in Sens protein output due to miR-9a repression of *sens* mRNA, we compared the ratio of mCherry-Sens to sfGFP-Sens in the following genotypes:

 1. Only *mCherry-sens* resistant to miR-9a repression

$$\frac{mCherry - sens^{m1m2}}{sfGFP - sens}; \frac{sens^{E1}}{sens^{E2}}$$

2. Neither *mCherry-sens* or *sfGFP-sens* resistant to miR-9a repression

$$\frac{mCherry - sens}{sfGFP - sens}; \frac{sens^{E1}}{sens^{E2}}$$

3. Only *sfGFP-sens* resistant to miR-9a repression

$$\frac{mCherry - sens}{sfGFP - sens^{m1m2}}; \frac{sens^{E1}}{sens^{E2}}$$

Single cell fluorescence values were obtained after cell segmentation and background subtraction as described earlier. Cells from individual discs were pooled together and red-green fluorescence was linearly correlated using least squares fit (QR factorization) to determine a slope and intercept for each disc. Next the average slope was calculated for each genotype (shown above). Fold reduction in mCherry-Sens protein output due to miR-9a was calculated as the ratio of slope-(1) to slope-(2) with relative errors propagated. Similarly, fold reduction in sfGFP-Sens protein output due to miR-9a was calculated as the ratio of slope-(2) to slope-(3).

## Topological domain structure

Heat maps of aggregate Hi-C data were used to calculate chromosomal contact frequency for embryonic nc14 datasets (*Stadler et al., 2017*) for landing sites at 22A3 and 57F5. DNase accessibility data (*Li et al., 2008*) and ChIP-seq of the insulator proteins CP190, BEAF-32, dCTCF, GAF and mod(mdg4) (*Nègre et al., 2010*) for the corresponding coordinates were analyzed as well.

## Experimental estimation of rate constants

### mRNA decay rate $D_m$

Female pre-pupal wing discs were dissected in WM1 medium (*Restrepo et al., 2016*) at room temperature. To inhibit RNA synthesis, discs were incubated in WM1 plus 5 µg/ml Actinomycin D in light protected 24-well dishes at room temperature. Approximately 20 discs were collected at 0, 10, 20 or 30 minutes post-treatment and were homogenized with 300 µl Trizol for RNA extraction and RT-qPCR analysis. Long-lived Rpl21 mRNA was used to normalize mRNA levels across time points. Similar results were obtained when 18S rRNA was used for normalization. mRNA decay was assumed exponential and a curve fit across all time-points was used to calculate the decay constant $D_m$ to be 0.0462 mRNA/min corresponding to a half-life $t_{1/2} = 15.75$ minutes ($R^2 = 0.91$). Hsp70 mRNA decay was also measured as an additional short-lived control with known half-life (observed $t_{1/2} = 35$ mins). All qPCR primers used are listed in the Key Resources Table.

### Protein decay rate $D_p$

Homozygous *3xFlag-TEV-StrepII-sfGFP-FlAsH-sens* (in a *sens* mutant background) female pre-pupal wing discs were dissected in WM1 medium at room temperature. Discs were incubated in WM1 plus 100 µg/ml cycloheximide for 0, 1, 2 and 3 hours at room temperature. Ten discs were harvested at each time-point and snap frozen in liquid nitrogen. To assay Sens protein abundance, we used an indirect sandwich ELISA (enzyme-linked immunosorbent assay) protocol as follows. Frozen discs were homogenized in 150 µl PBS containing 1% Triton-X, centrifuged to remove crude particulate matter and then incubated with rabbit anti-GFP (1:5000) overnight at 4°C in anti-Flag antibody coated wells. Wells were washed with PBS with 0.2% Tween 20 and incubated with HRP linked goat anti-rabbit (1:5000) antibody for 2 hours at 37C. Wells were subsequently washed and incubated with 100µl 1-Step Ultra TMB-ELISA substrate. HRP activity was terminated after 30 minutes with 100µl 2M $H_2SO_4$ and absorbance measured at 450 nm. Protein decay was assumed exponential and a curve fit across all time-points was used to estimate the decay constant $D_p$ to be 0.12 proteins/hr, corresponding to $t_{1/2} = 5.09$ hours ($R^2 = 0.84$).

## Protein synthesis rate $S_p$

As has been theorized previously (*Paulsson, 2005*; *Thattai and van Oudenaarden, 2001*) and also suggested by our experimental data (*Figure 4*), a constant Fano factor is related to the translation burst size b as follows

$$Fano\,factor = \left(\frac{\sigma^2}{\mu}\right) = 1 + b$$

Here $b$ is defined by the post-transcriptional rate constants as:

$$b = \left(\frac{S_p}{D_m + D_p}\right)$$

The Fano factor in the constant regime for Sens is ~ 20 molecules (*Figure 4D*). This is the cumulative Fano factor due to expression stochasticity of two alleles. Since variance is additive, Fano factor due to expression stochasticity of a single allele is ~10 molecules. Therefore, assuming b = (10-1) = 9 molecules and substituting the measured values for $D_m$ and $D_p$, we estimate that $S_p$ is ~ 0.5 proteins/mRNA/min. When miR-9a binding sites are deleted from the gene, Sens protein output is 1.80±0.21 fold higher. This makes the resistant protein synthesis rate $S_P$ ~ 1 proteins/ mRNA/min. and the Fano factor contribution from a single allele ~16.2 molecules. Thus, the cumulative Fano factor is ~ 35 molecules, as measured in experiments (*Figure 4D, E*). Thus, we fixed $S_p$ at 0.5 or 1 to simulate *sens* alleles with and without miR-9a binding sites respectively.

## Stochastic simulation model

We modeled the various steps of gene expression, based on central dogma, as linear first order reactions (*Figure 3—figure supplement 1A*). To simulate the stochastic nature of reactions, we implemented the model as a Markov process using Gillespie's Stochastic Simulation Algorithm (SSA) (*Gillespie, 1977*). A Markov process is a memoryless random process such that the next state is only dependent on the current state and not on past states. Simple Markov processes can be analyzed using a chemical master equation to provide a full probability distribution of states as they evolve through time. The master equation defining our three-variable gene expression Markov process is as follows:

$$\partial P(n_P, n_M, n_G, t)/\partial t$$
$$= S_m[P(n_P, n_M + 1, n_G, t) - P(n_P, n_M, n_G, t)]$$
$$+ D_m[(n_M + 1)P(n_P, n_M + 1, n_G, t) - n_M P(n_P, n_M, n_G, t)]$$
$$+ S_p n_M[P(n_P - 1, n_M, n_G, t) - P(n_P, n_M, n_G, t)]$$
$$+ D_p[(n_P + 1)P(n_P + 1, n_M, n_G, t) - n_P P(n_P, n_M, n_G, t)]$$
$$+ k_{on}[(n_{G_{total}} - n_G + 1)P(n_P, n_M, n_G - 1, t) - (n_{G_{total}} - n_G)P(n_P, n_M, n_G, t)]$$
$$+ k_{off}[(n_G + 1)P(n_P, n_M, n_G + 1, t) - (n_G)P(n_P, n_M, n_G, t)]$$

Here $n_P$ and $n_M$ denote the number of protein and mRNA molecules respectively. $n_{G_{total}}$ is the total number of genes of which $n_G$ are genes in the 'ON' state capable of transcription. Therefore, $n_G/n_{G_{total}}$ is the fraction of active genes. Time is denoted by $t$. The rate constants are defined in *Figure 3—source data 1*.

As the Markov process gets more complex, the master equation can become too complicated to solve. Gillespie's SSA is a statistically exact method which generates a probability distribution identical to the solution of the corresponding master equation given that a large number of simulations are realized.

## Simulation set-up and algorithm

The gene expression model is comprised of six events (*Figure 3—figure supplement 1A*) and their associated reaction rates. Unless specified, the events and rate constants were kept identical between *sfGFP-sens* and *mCherry-sens* alleles simulated in the same cell. At any given instance, for a given allele, either of these six events could take place.

Gillepsie's SSA is based on the fact that the time interval between successive events can be drawn from an exponential distribution with mean $1/r_{total}$ where

$$r_{total} = \sum_i r_i$$

That is the sum total of reaction rates for all $i$ events. Further, the identity of the event that will occur is drawn from a point probability defined as

$$P(i) = \frac{r_i}{r_{total}}$$

The algorithm proceeded as follows:

1. We initialized all simulations to start with no mRNA or protein molecules and promoter state set to 'off'.
2. $r_{total}$ was determined by calculating the individual rates $r_i$ at current time $t$ which depend on the number of substrate molecules and the rate constants in *Figure 3—source data 1*.
3. A random time interval τ was picked from the exponential distribution with mean $1/r_{total}$
4. A random event $i$ was picked with probability $P(i)$ as described above.
5. The cellular state was changed in accordance with the chosen event. The possible state changes were as follows

   a. Promoter state from off → on
   b. Promoter state from on → off
   c. mRNA molecule count increased by 1
   d. mRNA molecule count decreased by 1
   e. protein molecule count increased by 1
   f. protein molecule count decreased by 1
6. Simulation time was updated as $t + \tau$
7. Steps 2 to 6 were iterated until total simulation time reached 5 hr.

## Fano factor calculation

We ran simulations for 5 hr to approximate steady state expression, at the end of which protein and mRNA molecules produced from each simulation were counted. Simulations were randomly paired to mimic independent alleles within the same cell. A minimum of 5000 such simulation pairs were generated for each set of parameter values. For simulations that tested the effect of parameter gradients on Sens noise, we varied the relevant parameter across a defined range, with 20 evenly spaced values comprising the sweep. Each parameter value was used to make paired simulations as above, after which paired simulations from the entire sweep were pooled to generate a whole population. This population was binned into 25–30 bins based on total Sens number per pair, and the Fano factor was calculated for each bin. Bootstrap with resampling was used to determine 95% confidence intervals for each bin's Fano factor.

## Parameter constraints

To keep simulations computationally feasible, we adjusted the slowest rate parameter, the protein decay rate $D_p$, from 0.002 proteins/min to 0.01 proteins/min (half-life from 5 hours to 1 hour). This is because we conducted simulations until protein conditions reached steady state, which is approximately five-fold longer than the half-life for the slowest reaction. For 25-hour simulations, this was resource and time-intensive. We compared the noise trends in simulations with either $D_p$ of 0.002 proteins/min or to 0.01 proteins/min, and found both generated similar noise trends to one another. This indicates that protein decay is a not a major source of intrinsic noise in this model. Therefore, we kept $D_p$ at 0.01 proteins/min.

The transcriptional parameters $S_m$, $k_{on}$ and $k_{off}$ were varied in accordance with the specific hypothesis being tested. We constrained them loosely to be within an order of magnitude of reported values for these rates from the literature (*Milo et al., 2010*). We also constrained these rates so as to produce steady state protein numbers and Fano factors similar to experimental data. The minimum and maximum values used are listed in *Figure 3—source data 1*.

## Modeling sens regulation by Wg signaling

As seen in *Figure 2A*, Sens-positive cells display a wide range of expression and they are patterned in space as stripes. This is due to signaling via Wg, which is secreted from the presumptive wing margin and diffuses to form a bidirectional gradient. Wg signaling directly activates transcription of the *sens* gene (*Eivers et al., 2009*; *Jafar-Nejad et al., 2006*). We assumed that at least one of the three transcriptional rate parameters ($S_m$, $k_{on}$ or $k_{off}$) in our model must be responsive to Wg signaling. We systematically varied one of the parameters while keeping the others constant. In all cases, varying the parameter did produce a spectrum of Sens expression levels (*Figure 3*).

We next calculated the Fano profile for each case. Only a free variation in $k_{on}$ produced a Fano profile that resembled the experimental data, with a Fano peak at the lowest Sens levels which dramatically declines as Sens levels increase (*Figure 3E*). Thus, to recreate a Sens gradient *in silico* we kept $S_m$, $D_m$, $S_p$, $D_p$ and $k_{off}$ constant, and we varied $k_{on}$ from 0.025 to 10 min$^{-1}$. Since $1/k_{on}$ defines the average time the promoter is inactive, this varied from 6 seconds to 40 minutes in our model.

## Impact of transcription burst kinetics

Given that average time the promoter is 'off' is $1/k_{on}$ and average time it is 'on' is $1/k_{off}$, we define transcription burst size and burst frequency as follows

$$BurstSize = \left(\frac{S_m}{k_{off}}\right)$$

$$BurstFrequency = \left(\frac{1}{k_{on}} + \frac{1}{k_{off}}\right)^{-1}$$

It is worth noting these values define the average burst size or frequency across exponentially distributed values. We independently varied burst size with $S_m$ (*Figure 3D*) and burst frequency with $k_{on}$ (*Figure 3E*).

As described previously, a gradient in $k_{on}$ can re-create the experimentally observed noise profile. Together, these observations suggest that perhaps the Wg gradient translates into a gradient of *sens* promoter burst frequencies - at low Wg concentrations, burst frequency is low and at high concentration, the promoter switches states rapidly. In general, we found that as promoter state switching time-scales get smaller with respect to mRNA or protein lifetimes, bursting dynamics negligibly contribute to expression stochasticity (*Figure 3E*). This is expected since frequent individual transcription bursts get time-averaged on the scale of long lived mRNA or proteins (*Paulsson, 2005*). From above, it is clear that either $k_{on}$ or $k_{off}$ could be rate-limiting to determine burst frequency. Therefore, we also tested the effect of only varying $k_{off}$ while keeping the other 5 parameters constant ($k_{on} = 1$/min i.e. non-limiting). Interestingly, a gradient of $k_{off}$ produced a very distinct Fano profile that peaked at approximately half-maximal protein expression (*Figure 3F*). $k_{off}$ is a coupled parameter that simultaneously affects both transcription burst size and frequency.

After recreating the graded expression of Sens, we next sought to understand which burst parameter(s) could explain the effect of genomic position on Fano factor. Modulating burst frequency simply regenerated the noise profile seen before, as expected. Increasing burst size with $S_m$ (or even with the coupled parameter $k_{off}$) mimicked the higher and larger Fano peak change as seen for *sens* at 57F5/57F5 (*Figure 6C*). To simulate altered *cis*-regulation of the 57F5 allele, we simulated cells with two *sens* alleles at 57F5/22A3, applying two different $S_m$ values corresponding to a burst size of either 4 or 8 mRNAs. As before, alleles were simulated independent of each other to generate the Fano factor profile (*Figure 6—figure supplement 2A*). To simulate *trans* allelic inhibition in 57F5/57F5 cells, we set burst size of 4 mRNAs for both alleles but decreased the $k_{on}$ value of one allele if the other was in the 'on' state (*Figure 6—figure supplement 2B*). As shown, *trans* allelic inhibition led to greater noise between the alleles and a higher Fano peak. Yet, even at 90% inhibition (*Figure 6—figure supplement 2B*) it did not affect peak width as observed experimentally. Therefore, while promoter competition might be occurring, it alone does not explain our results.

## Relationship between protein level and 'constant' Fano factor

If $k_{on}$ and $k_{off}$ are not limiting so that the promoter is in the ON state 100% of the time, the steady state protein level is described as:

$$Protein = \left(\frac{S_m S_p}{D_m D_p}\right)$$

Thus, once the promoter is fully occupied, protein expression must be increased by regulating the birth-death rate constants. Correspondingly, the Fano factor will be:

$$Fano\ factor = 1 + b$$
$$= 1 + \left(\frac{S_p}{D_m + D_p}\right)$$

If $b \gg 1$, then we have:

$$Fano\ factor \sim \left(\frac{S_p}{D_m + D_p}\right)$$

Thus the Fano factor must rise with protein level if these rate constants are perturbed. When we freely vary $S_p$, $D_m$ or $D_p$ in simulations, we recreate this linear relationship such that if the rate constant is biased towards greater Sens protein accumulation, the corresponding Fano factor increases (*Figure 3—figure supplement 1D*). We also observe signatures of a slowly rising Fano factor in our data in the regime we describe as 'constant' Fano noise. We therefore speculate that $S_p$, $D_m$ or $D_p$ might vary across the developmental field to expand the range of steady state Sens accumulation independent of the *sens* promoter.

## Quantification and statistical analysis

Only numbers of cells identified as 'Sens-positive' were used for the data analysis. We identified N = 50,788 Sens-positive cells that were singly-tagged wildtype *sfGFP-sens* and *mCherry-sens* alleles inserted in locus 22A3. For the miR-9a binding site mutant *sens* allele pair at locus 22A3 we identified N = 106,738 cells. Tandem tagged *sfGFP-mCherry-sens* data was gathered for N = 24,970 cells pooled from both loci since cells from individual loci produced identical Fano factor profiles. For the repression assay, we measured genotypes with both alleles wildtype for N = 20,353 cells; mCherry tagged *sens* wildtype paired with a mutant sfGFP tagged *sens* for N = 19,088 cells, and sfGFP tagged *sens* wildtype paired with a mutant mCherry tagged *sens* for N = 12,947 cells. A second set of repression measurements shown in *Figure 4B* was done for both alleles mutant (N = 6369 cells) or wild type *mCherry-sens* paired with mutant *sfGFP-sens* (N = 7557 cells). Fano factor profiles comparing the effect of genomic locus contain data from N = 19,549 cells (22A3/22A3), N = 27,221 cells (57F5/57F5) and N = 13,776 cells (22A3/57F5).

Due to the nature of the experiments, there is no technical replication. Rather, the numbers listed above refer to each Sens-positive cell as a biological replicate. Since ~1,000 cells could be measured in one wing disc sample, the experiments utilized 12–100 wing disc biological replicates per genotype.

For ectopic mechanosensory bristle measurements, we counted the frequency of adult wings with at least one ectopic sensory organ. The number of adult wings analyzed per genotype ranged from 60 to 88. Each wing is considered a biological replicate. For parent stocks containing the ψC31 docking site only and no *sens* transgene, the following frequencies were observed: N = 0/60 wings (parent-22A3/22A3 and parent-22A3/57F5) and N = 0/88 wings (parent 5757/57F5). For transgenic stocks with wildtype *sens* we observed N = 2/62 wings (*sens* [+ miR-9a] - 22A3/22A3), N = 1/60 wings (*sens* [+ miR-9a] - 22A3/57F5) and N = 23/79 wings (*sens* [+ miR-9a] - 5757/57F5). For transgenic stocks with mutant *sens* we observed N = 3/83 wings (*sens* [- miR-9a] - 22A3/22A3), N = 4/64 wings (*sens* [- miR-9a] - 22A3/57F5) and N = 12/65 wings (*sens* [- miR-9a] - 5757/57F5).

The 95% confidence intervals for all point estimates (mean, Fano factor) of Sens protein data and model simulations were built by bootstrapping with resampling within individual binned cell populations. In all cases, the distribution generated by bootstrapping was checked for normality before obtaining the 2.5$^{th}$ and 97.5$^{th}$ percentile values. The error frequencies of adult bristle patterns were

statistically tested by calculating the odds ratio of error frequency between pairs of genotypes. A Fischer's exact test was applied to determine if the odds ratio significantly deviated from one.

For ectopic chemosensory bristle measurements, we measured the average distance between pairs of neighboring bristles. N = 174 bristle neighbors (parent-22A3), N = 177 bristle neighbors (parent-57F5), N = 1004 bristle neighbors (wild type *sens* at 22A3), N = 1063 bristle neighbors (wild type *sens* at 57F5), N = 944 bristle neighbors (mutant *sens* at 22A3) and N = 605 bristle neighbors (mutant *sens* at 57F5). Each pair of neighboring bristles comprises one biological replicate. 95% confidence intervals were calculated by bootstrapping empirical distributions of bristle pair distances across genotypes, and these were compared statistically using a student's t-test. All other statistical tests and image quantification procedures are listed in the corresponding methods and figure legends sections.

There was no exclusion of any data or subjects.

## Acknowledgements

Fly stocks from Hugo Bellen and the Bloomington*Drosophila*Stock Center are gratefully appreciated. Antibodies were gifts from Hugo Bellen and purchases from the Developmental Studies Hybridoma Bank. We thank Koen Venken for extensive help with BAC recombineering protocols and reagents. We thank Michael Stadler and Michael Eisen for generously providing HiC maps of the 22A3 and 57F5 loci from their studies. We thank Jessica Hornick and the Biological Imaging Facility for help with imaging. Financial support was provided from the Northwestern Data Science Initiative (RG), Robert H Lurie Comprehensive Cancer Center (RG), Pew Latin American Fellows Program (DMP), Max Planck Society (DKP and PT), Chancellor's Fellowship of University of Edinburgh (DKP), NIH (R35GM118144, RWC), NSF (1764421, MM and RWC), and the Simons Foundation (597491, MM and RWC). MM is a Simons Foundation Investigator.

## Additional information

### Funding

| Funder | Grant reference number | Author |
| --- | --- | --- |
| National Institutes of Health | R35GM118144 | Ritika Giri<br>Diana M Posadas<br>Hemanth K Potluri<br>Richard W Carthew |
| Simons Foundation | 597491 | Madhav Mani<br>Richard W Carthew |
| National Science Foundation | 1764421 | Madhav Mani<br>Richard W Carthew |
| Pew Charitable Trusts | Pew Latin American Fellows Program | Diana M Posadas |
| Max Planck Society | MPI Funding | Dimitrios K Papadopoulos<br>Pavel Tomancak |
| Northwestern University | Data Science Initiative | Ritika Giri |
| Robert H Lurie Comprehensive Cancer Center | | Ritika Giri |

The funders had no role in study design, data collection and interpretation, or the decision to submit the work for publication.

### Author contributions

Ritika Giri, Conceptualization, Resources, Data curation, Software, Formal analysis, Investigation, Visualization, Methodology; Dimitrios K Papadopoulos, Resources, Data curation, Formal analysis; Diana M Posadas, Hemanth K Potluri, Resources; Pavel Tomancak, Supervision; Madhav Mani, Conceptualization, Supervision, Investigation, Methodology, Project administration; Richard W Carthew, Conceptualization, Formal analysis, Supervision, Funding acquisition, Project administration

## Author ORCIDs

Ritika Giri (iD) http://orcid.org/0000-0001-8838-0818
Pavel Tomancak (iD) http://orcid.org/0000-0002-2222-9370
Richard W Carthew (iD) https://orcid.org/0000-0003-0343-0156

## Decision letter and Author response

Decision letter https://doi.org/10.7554/eLife.53638.sa1
Author response https://doi.org/10.7554/eLife.53638.sa2

# Additional files

## Supplementary files

• Transparent reporting form

## Data availability

All data generated and analyzed during this study are included in the manuscript and supporting files. Source data is provided for all main figures and computer code for analysis and modeling are available at https://github.com/ritika-giri/stochastic-noise (copy archived at https://github.com/eli-fesciences-publications/stochastic-noise).

The following previously published datasets were used:

| Author(s) | Year | Dataset title | Dataset URL | Database and Identifier |
|---|---|---|---|---|
| Stadler MR, Haines JE, Eisen MB | 2017 | Convergence of topological domain boundaries, insulators, and polytene interbands revealed by high-resolution mapping of chromatin contacts in the early Drosophila melanogaster embryo | https://www.ncbi.nlm.nih.gov/geo/query/acc.cgi?acc=GSE100370 | NCBI Gene Expression Omnibus, GSE100370 |
| Shah PK, Kheradpour P, Morrison CA, Henikoff JG, Feng X, Ahmad K, Russell S, White RAH | 2010 | A comprehensive map of insulator elements for the Drosophila genome | https://www.ncbi.nlm.nih.gov/geo/query/acc.cgi?acc=GSE16245 | NCBI Gene Expression Omnibus, GSE16245 |

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
