## [Decision Letter]

Thank you for submitting your article "Ordered patterning of the sensory system is susceptible to stochastic features of gene expression" for consideration by *eLife*. Your article has been reviewed by three peer reviewers, one of whom is a member of our Board of Reviewing Editors, and the evaluation has been overseen by Patricia Wittkopp as the Senior Editor. The reviewers have opted to remain anonymous.

Generally, the reviewers agreed this is a very interesting paper dealing with a very important topic. As you will see below, reviewer #1 and #3 only request a number of editorial changes which are important and straight-forward to implement.

Reviewer #2 has been much more critical, raising a total of nine points. In ensuing discussions, reviewer #1 and #3 have looked over these nine points and engaged with reviewer #2 in discussions about what exactly should be done in regard to these nine points. Here are the conclusions that we hope you can all address. The numbering matches the numbering of reviewer #2:

1) Transcript counting: We do not request that you do mRNA measurements. However, please do discuss either the Result or Discussion considerations about mRNA vs protein measurements. One reviewer notes that there is an extensive literature of using protein measurements coupled with modeling to deduce transcriptional bursting parameters and, from one reviewer's perspective, when mRNA FISH and MS2 measurements when subsequently done on systems that had previously had only protein reporter measurements, the RNA data on the whole supported what was deduced from protein reporter measurements.

2) Automated cell ID approach: All reviewers agree that you need to provide a validation of your automated cell ID system comparing manual and automated scoring.

3) Indeed, please explain the total molecule counts between these two experiments (Figure 2). If the dual reporter is not double the single reporter, the authors should provide an explanation. Biologically, I think this is a very important point for their system and should be addressed clearly and directly.

4) Indeed, please explain the quantitative differences between the data and their model.

5) The authors argue that Sens levels are the same at site 22A3 and 57F5: This is not clear. The levels look quite different. This is critical for many of the authors' conclusions, especially that the phenotypic differences are a result of noise, not levels. The authors must provide a compelling quantitative argument that these levels are the same.

6) The HiC data should be moved to the Supplement.

7/8) No experimentation required. Reviewer #1 and reviewer #3 – and then eventually also reviewer #2 – agreed that the most important thing is here is a clarification of the limitations of the assay. Please add sentences to address the ectopic reporter and correlation concerns.

9) Please address the point raised by reviewer #2.

Again, as stated, above those are the nine points raised by reviewer #2. Please also address the points raised by reviewer #1 and reviewer #3.

Reviewer #1:

This is an interesting and technically accomplished study from the Carthew lab. The paper shows, using protein count estimates in live embryos, that transcriptional bursts can have an effect on protein level biology, that expression noise can be sensitive to chromosome position (even in contexts where the protein is effectively functional. The study then shows that somatic pairing effects can affect protein noise, but these effects depend again on the genome context. Finally the paper shows how an enhanced level of noise can affect the lateral inhibition process. My main critique about the paper is related more to the way it is presented:

"We contribute to this effort by providing the first study on the impact of stochastic expression in a developmental process involving complex cell-cell signaling. "

This is overstated, there are several other examples where this issue has been directly addressed in embryos, for example, Raj et al., 2010 and Lagha et al. Cell. 2013 153:976-87. In any kind of tissue, there is complex cell-cell signalling, even in an otherwise cell-autonomous specification system (which you could argue the bristles are, at least from the perspective of initiating the decision)

"However, the process of cell differentiation frequently begins with a fate-determining protein expressed transiently at low levels, and expression then either greatly increases or decreases, corresponding to divergent fate adoption".

I think this may also be misleading, especially in developmental contexts. Accurate quantitation in the cells making the decisions is comparatively rare. I know there is the urban myth that TFs are low abundance, but there are many contradictory examples. Please feel free to cite something that proves me wrong.

I'm not sure about Figures 1D and 1E. It may just be the language used, but why would deterministic gene expression follow a single line, is it even useful to make this distinction here. 1E looks more like an Elowitz intrinsic vs. extrinsic noise plot. It is just confusing to change the language, unless this is adequately justified, and there is little in the text around these figures to justify this. The language becomes easier later on, when dealing with the Fano factor.

Subsection “Counting Sens Proteins to Measure Expression Noise”- in the double labeled line, is the fly still mutant for the endogenous *sens* allele? Please make this clear to the reader.

Figure 2: measuring the technical contribution using the double tag, does this involve a number of assumptions such as equivalence in turnover times, folding times etc.? Or by technical, do you just mean any removing any non-linearity in microscope detection? These things should be more openly discussed.

Figure 3: this is an unusual formulation for the burst frequency. It is usually expressed as just the k_on_, or k_on_.τ, where τis the RNA lifetime. Please justify this. Does this matter in your later inferences, for example, subsection “Sens Protein Noise Displays a Signature Arising from Transcription Bursts”.

Discussion paragraph two. I think the implied argument that pairing is important in humans should be softened. Somatic pairing effects may have been picked up, but this is far from mainstream. The standard view is more along the lines of stochastic repositioning of chromosomes with respect to each other and nuclear compartments each cell division. Yes, there may be opportunistic interactions forming between accidentally opposed loci, but I would avoid making too much of these anecdotal papers on pairing in mammalian cells (which were all before the large scale expansion in field using current methods of studying chromosome organization).

References to Nanog should probably be removed. in vivo, Nanog does not fluctuate much. It turns on, stays on for a couple of days, then turns off (Hadjantonakis lab). Most of the culture studies also see very slow fluctuations (6-7 cell divisions before a high cell will revert to the mean) which is longer than the gene is on in vivo.

Reviewer #2:

In this paper, the authors investigated how noise in the expression of the transcription factor Sens affects sensory bristle patterning of the fly wing. They generated tagged *sens* BAC transgenes labeled with scGFP or mCherry. They evaluated noise from these two alleles inserted into the 22A3 insert site. Since these transgenes are translational fusions that include the Sens protein, the noise could arise from transcription and/or translation. They examine expression when miRNA binding sites are knocked out, and find that noise increased as predicted if the main source of noise was transcription. They then examine expression at the 57F6. They find similar noise if there is a single copy at 57F6 and a single copy at 22A3. However, they find a different noise pattern if there are two copies at 57F6, which they argue is evidence that allele pairing and transvection generates noise at this locus. They then examine expression in the wing and argue that the levels are similar between transgenes inserted at 22A3 and 57F6, but the noise is different. Flies with two copies of 57F6 have wing phenotypes and the authors argue that this is due to the change in noise.

This is a very interesting topic. However, there are many technical and conceptual issues with the paper.

Major Comments

1) Translational fusion reporter genes complicate the conclusion that transcription is the source of noise: It is not clear why the authors examined protein noise. This complicates the system greatly (see points below), as they are looking downstream of transcription. The authors should examine transcription directly by conducting either 1. RNA FISH on GFP and cherry and evaluating variability in their experimental conditions, and/or 2. Generate transcriptional reporters and examine expression.

2) The authors have not validated their automated cell ID approach: The efficiency and accuracy of the automated system is not reported. The authors should validate the cell IDs manually and report the accuracy including the percentage of false positives and false negatives.

3) The dual tag experiment in Figure 2B should have twice as many molecules of GFP and mCherry as in Figure 2A. The authors compare singly tagged GFP and mCherry reporters to doubly tagged reporters. Their results suggest that the total number of molecules is equivalent (Compare Figure 2B to 2A). However, the number of molecules for the double tag should be double that of the singly tagged reporters. This result suggests that there are major issues with the cell ID, expression quantification, and/or analysis. Alternatively, these results could be explained if the reporters hit a biological maximum for these molecules. This possibility is also a concern. The authors must explain this result.

4) The model does not match the data: The authors suggest that the model presented in Figure 3C, 3E matches their data best. However, the absolute quantities of molecules does not match the data in Figure 2C. The authors should present how well the data fits their model.

5) The authors argue that Sens levels are the same at site 22A3 and 57F5: This is not clear. The levels look quite different. This is critical for many of the authors' conclusions, especially that the phenotypic differences are a result of noise, not levels. The authors must provide a compelling quantitative argument that these levels are the same.

6) The TAD and chromatin is overinterpreted and unnecessary: The authors provide analysis in Figure 6 to show that the two insert sites are different. These data do not make a compelling argument, the analysis is incomplete, and the conclusion is fairly obvious. The authors argue that their analysis shows that the two insert sites have different chromatin environments. Wouldn't this be true of any two sites in the genome? The authors suggest that the TADs are different, but they do not conduct a proper analysis. The authors should provide and examine directionality indices to make their TAD calls. Also, the HiC is from embryos. Though TADs are generally similar across tissues, this is not absolute. For the authors to make this point, they should conduct hiC on wing discs. In general, this section does not add to the paper. The conclusion that insert sites have different chromatin environments is generally agreed upon.

7) The authors do not conduct an in-depth analysis of pairing or transvection: The authors conclude that interactions/transvection between the two *sens* alleles at 57F5 cause the increase in noise. However, there are problems with this argument. First, do the loci pair differently at 22A3 and 57F5? The authors should conduct DNA FISH at the site with and without the transgene to answer this question. Second, do these sites pair/loop to the other insert site and/or endogenous *sens*? The authors should conduct DNA FISH with and without the transgenes to test for these chromatin interactions.

8) The authors conclude that transvection increases noise, yet these experiments are completely heterologous: The authors examine transvection and noise at two sites and make their conclusions. There are problems with this rationale. First, is this a general principle for *sens*? Which is the general rule: transvection independent noise (as seen at 22A3) or transvection dependent noise (as seen at 57F5)? The authors should conduct these experiments at several additional locations to answer this question. Also, variable transvection at different sites has been described (ex: King, et al., 2019).

Second, what does this conclusion mean/why is it important? It is well known that transgenes can cause strange effects dependent on position. To address this issue, the authors should use CRISPR to insert reporters into endogenous *sens* and examine noise. However, I'm still not sure what conclusion about biology can be concluded from the transgene experiments.

Third, the differences at 22A3 and 57F5 could be due to local transcription changes (aka chromatin) or local pairing/transvection differences. The authors say that local pairing/transvection differences drive the difference but provide no evidence. To address this issue, the authors should test the transvection ability of each locus using canonical transvection assays involving the white or yellow genes.

9) The authors argue that noise drives the phenotypic differences at 22A3 and 57F5 yet the protein levels are the same – this is not a coherent argument: In Figure 7, the authors argue that the levels of Sens protein are the same at 22A3 and 57F5. This is not convincing. The authors should provide a quantitative analysis of the position and expression of Sens in these cells. The position is critical. For example, the central cells could be higher and outer cells could be lower for 22A3 compared to 57F5. This would cause the similarity in quantification of total expression seen in Figure 5C (which is not convincing, as discussed above), yet the spatial differences could cause the phenotype.

A bigger issue is the argument that noise drives the phenotype. If the protein levels are identical, it should not matter which allele provides the protein. In other words, if there needs to be 10 units of Sens in a cell, it does not matter if the less noisy 22A3 provide 5 and 5 from each allele whereas the noisier 57F5 provides 4 and 6 from each allele (or 3 and 7, etc.). At the end of the day, the absolute protein quantity should drive the phenotype, not the ratio of protein generated from each allele. The only exception would be if the alleles were different in some way. The only source of difference here would be the tags. If the tags are generating the phenotype, this is also problematic.

The authors must provide an explanation to justify this main conclusion of their paper.

Reviewer #3:

Overall, we found the manuscript by Giri et al. to be interesting and exceptionally well done, one of the cleanest analyses in the noise field in some time. The only substantial critique that one can make is the lack of single-cell RNA measurements of noise (e.g., by single-molecule RNA FISH) to validate the modeling predictions which are based on single-cell protein noise measurements. However previous studies in the field have also relied only on protein measurements (see below) and in the spirit of e*Life*, we feel this validation can be left for future work as long as it is stated in the Discussion as a basis for future work.

The comments we present below are intended solely to improve readability, help support the authors' claims, and avoid potential confusion.

Major comments:

1) In the last paragraph of subsection “Allele Pairing at 57F5 Generates *Trans* Regulation and Enhanced Noise” and Discussion paragraph two, the authors claim that the noise peak for paired *sens* alleles at 57F5 results from altered bursting kinetics, specifically an enhanced burst size. Although their modelling supports this claim, there is no direct measurement of RNA bursting kinetics through techniques such as single molecule RNA-FISH. Previous results support the hypothesis of transcriptional burst size modulation (PMID: 24903562) but the authors should also address alternate potential mechanisms (i.e., PMIDs 27153498, 26760529, 26544860), for example the role of alternative splicing (PMIDs: 29986741, 31222776), even if for contrast, as such mechanisms may not be functioning for *sens*. It should be made clear in the Discussion section that the lack of mRNA quantification is a limitation, and despite existing precedent for burst size modulation, such RNA measurements will be important experiments in future work.

Substantive remarks:

1) In Figures 4B and 5D the authors have overlaid scatterplots from separate measurements. In regions of high density, there is a loss of information as to where the center of mass lies. The authors should consider using a contour plot or shading to convey density.

2) Introduction paragraph three the authors list nuclear retention of transcripts as a mechanism for reducing protein noise that may arise from transcriptional bursts. We suggest the authors exercise caution here as the cited papers did not measure protein noise and there is now direct competing evidence indicating that nuclear export amplifies RNA/protein noise in the cytoplasm (PMIDs: 30243562, 30359620). Some might argue that the evidence in this report (increasing protein Fano factor) contradicts the papers cited which claim that nuclear export attenuates noise from transcriptional bursts to minimal Poisson levels.

3) In the final paragraph, the authors may want to mention and cite the evidence of other examples where stochastic transcriptional fluctuations appear to have evolved as a mechanism for influencing cell fate (PMIDs: 17379809, 16051143, 28607484).

---

## [Author Response]

Reviewer #1:"We contribute to this effort by providing the first study on the impact of stochastic expression in a developmental process involving complex cell-cell signaling. "This is overstated, there are several other examples where this issue has been directly addressed in embryos, for example, Raj et al., 2010 and Lagha et al. Cell. 2013 153:976-87. In any kind of tissue, there is complex cell-cell signalling, even in an otherwise cell-autonomous specification system (which you could argue the bristles are, at least from the perspective of initiating the decision)

We have deleted the sentence.

"However, the process of cell differentiation frequently begins with a fate-determining protein expressed transiently at low levels, and expression then either greatly increases or decreases, corresponding to divergent fate adoption".I think this may also be misleading, especially in developmental contexts. Accurate quantitation in the cells making the decisions is comparatively rare. I know there is the urban myth that TFs are low abundance, but there are many contradictory examples. Please feel free to cite something that proves me wrong.

We have deleted the sentence.

I'm not sure about Figures 1D and 1E. It may just be the language used, but why would deterministic gene expression follow a single line, is it even useful to make this distinction here. 1E looks more like an Elowitz intrinsic vs. extrinsic noise plot. It is just confusing to change the language, unless this is adequately justified, and there is little in the text around these figures to justify this. The language becomes easier later on, when dealing with the Fano factor.

We have removed the allusion to a deterministic behavior in Figure 1D and E. The plot in 1D only has the stochastic trace, and the straight line in 1E refers now to the moving average.

Subsection “Counting Sens Proteins to Measure Expression Noise”- in the double labeled line, is the fly still mutant for the endogenous sens allele? Please make this clear to the reader.

Yes it is and has been clarified. Added phrase “in an endogenous *sens* null background”

Figure 2: measuring the technical contribution using the double tag, does this involve a number of assumptions such as equivalence in turnover times, folding times etc.? Or by technical, do you just mean any removing any non-linearity in microscope detection? These things should be more openly discussed.

The technical contribution involves both detection of photons and also biological sources such as differences in turnover /denaturation of the FPs and differences in folding times. We have clarified this in the text of the Results and also the Experimental Materials and methods sections discussing technical noise.

Figure 3: this is an unusual formulation for the burst frequency. It is usually expressed as just the kon, or kon .τ, where τ is the RNA lifetime. Please justify this. Does this matter in your later inferences, for example, subsection “Sens Protein Noise Displays a Signature Arising from Transcription Bursts”.

Indeed, k_on_ is a reasonable approximation of burst frequency when K_off_ is much greater than k_on_. Under these circumstances, bursts are very short and so most time is spent with the promoter in the OFF state. Thus, k_on_ – conversion of OFF to ON, limits the frequency of bursts. However, when k_off_ is similar or smaller than k_on_, a significant amount of time is spent with the promoter in the ON state, where a new burst cannot yet occur. A more complete formulation of burst frequency is the one that we use. Note that when k_off_ is much greater than k_on_, our formulation simplifies to just k_on_. Since we sweep the k_on_ and k_off_ parameters without assuming one is necessarily greater than the other, our formulation is the most general form for the burst frequency.

Discussion paragraph two. I think the implied argument that pairing is important in humans should be softened. Somatic pairing effects may have been picked up, but this is far from mainstream. The standard view is more along the lines of stochastic repositioning of chromosomes with respect to each other and nuclear compartments each cell division. Yes, there may be opportunistic interactions forming between accidentally opposed loci, but I would avoid making too much of these anecdotal papers on pairing in mammalian cells (which were all before the large scale expansion in field using current methods of studying chromosome organization).

We have modified the sentence to eliminate reference to mice or humans.

References to Nanog should probably be removed. in vivo, Nanog does not fluctuate much. It turns on, stays on for a couple of days, then turns off (Hadjantonakis lab). Most of the culture studies also see very slow fluctuations (6-7 cell divisions before a high cell will revert to the mean) which is longer than the gene is on in vivo.

We have deleted the entire section and references related to Nanog in the Discussion.

Reviewer #2:1) Transcript counting: We do not request that you do mRNA measurements. However, please do discuss either the Result or Discussion considerations about mRNA vs protein measurements. One reviewer notes that there is an extensive literature of using protein measurements coupled with modeling to deduce transcriptional bursting parameters and, from one reviewer's perspective, when mRNA FISH and MS2 measurements when subsequently done on systems that had previously had only protein reporter measurements, the RNA data on the whole supported what was deduced from protein reporter measurements.

We were also concerned with the question of protein vs RNA measurements. Another student, Rachael Bakker, in the Carthew lab has been conducting her thesis research on adapting single molecule FISH (smFISH) for *Drosophila* imaginal discs, something that had not been done before. She has successfully developed the method and built an analytical pipeline to count mRNA number and nascent transcribing RNA in individual wing cells, across the tissue. Although she has used this to characterize *senseless* expression, she greatly expanded its use to study several genes responsive to Dpp signaling. The culmination of her work has now been prepared as a manuscript for publication. We decided not to include her *senseless* work in the protein noise manuscript because Rachael’s efforts have been outstanding and we did not want to diminish the impact of her complete study. Her manuscript has now been deposited in biorXiv (doi.org/10.1101/2020.01.24.918623) and we cite this preprint in the Results and Discussion, in two paragraphs addressing the concerns of the reviewers. We refer to the preprint providing RNA evidence to support *senseless* transcriptional regulation occurring via modulation of burst frequency. This was the effect predicted by the present protein study. Rachael also found that the Fano factor (noise) of *senseless* mRNA expression is greater than 1, an indicator of transcriptional bursting, which is a strong prediction of the present protein based study. In addition to our discussion of this preprint, we also discuss the papers that the reviewer alludes to.

One of the things we wanted to do with smFISH was compare *sens* mRNA and protein numbers in a cell at the same time. A simple correlation would strengthen the use of protein to infer transcription kinetics. However, the smFISH method we developed for imaginal discs has the unfortunate consequence of destroying protein epitopes and native GFP and mCherry fluorescence activities. So we could not measure both RNA and protein in the same sample. Altering the method to retain protein integrity results in high backgrounds for the smFISH signal.

A two-color smFISH assay probing GFP and mCherry tagged alleles in the same cell would be an exciting additional measurement and analysis to conduct. Although such a two-color assay is possible (see preprint for the experiment demonstrating it), the technical challenge of designing probes specific for GFP that do not recognize mCherry (and vice versa) has meant the signal of such hybridized probes is too weak in the 565nm channel to be reliable above background. Although we are still working on resolving the issue, we are not confident to definitively conclude a concordance between RNA and protein in our work. Therefore, the discussion also points out the limitations on not affirming the model by looking at intrinsic RNA noise and transcriptional dynamics more directly.

2) Automated cell ID approach: All reviewers agree that you need to provide a validation of your automated cell ID system comparing manual and automated scoring.

We have performed the validation as requested. The description of the validation has been inserted into the Materials and methods section discussing nuclear segmentation. Testing over 500 manually curated nuclei randomly chosen, the approach correctly identified 95.1% of manually curated nuclei, with 2.6% false positives and 2.3% false negatives.

3) Indeed, please explain the total molecule counts between these two experiments (Figure 2). If the dual reporter is not double the single reporter, the authors should provide an explanation. Biologically, I think this is a very important point for their system and should be addressed clearly and directly.

There are two copies of the tandem-tag gene in the experiments. We had not displayed the entirety of the XY axes in Figure 2A,B in order to keep the distribution of the tandem-tag within the same range as the single allele experiment. The protein levels in the tandem-tag cells do indeed extend higher than seen with single tags, as expected. The median expression is 362 ± 2 molecules for the single reporter and 767 ± 12 molecules for the tandem reporter. As expected, this corresponds to a fold increase of 2.1 ± 0.03. Further, as kindly pointed out by reviewer 3, the data are over-plotted and therefore an intuitive estimation is difficult. We modified Figure 2A,B and reproduce the scatter plot in its entire range. Colors represent number of cells present in each hexagonal region (12,000 cells from each dataset were sampled for cell density comparison).

4) Indeed, please explain the quantitative differences between the data and their model.

We thank the reviewer for pointing out this error. The simulations in Figure 3 were done with the translation rate *S*_p_ set to *sens (- miR-9a).* We have corrected Figure 3 by using the translation rate for *sens (+ miR-9a)* which is the genotype for Fano factor in Figure 2C. We have also enhanced results in Figure 3 by showing noise trends across a range of transcription parameters that were fixed for each sweep.

More broadly, the purpose of the modeling is to predict distinct qualitative trends in the data. We did not mean to imply that the model’s purpose is to find parameter values that quantitatively “best-fit” the data. We could do that by sweeping free parameter values and fitting simulations to data, and we would definitely find a parameter set that would best-fit. However, it is not the purpose of modeling in Figure 3 to find such values for k_on_, k_off_, *S*_m_. Since we cannot directly measure these rate constants experimentally, any value prediction would be futile. Moreover, freely varying multiple parameters to fit a model always works no matter what the model structure is. John von Neumann was quoted to say, “With four parameters I can fit an elephant, and with five I can make him wiggle his trunk.”

Instead, Figure 3 asks if regulation of *sens* expression acts on just one of the transcription rate constants, does the model predict qualitatively distinct profiles of Fano factor in protein data? The answer is yes: the noise profiles are qualitatively different from one another. We then compare the trends of each to the experimental data, and see that the scenario where k_on_ varies by regulation generates a noise profile that is most similar. Indeed, our smFISH preprint (Bakker et al., 2020) validates this conclusion using a completely independent means to test the hypothesis (see above). We have added text in the Results to more explicitly state our goals to qualitatively test and not quantitatively test, plus we refer to the preprint as further support for our conclusion.

5) The authors argue that Sens levels are the same at site 22A3 and 57F5: This is not clear. The levels look quite different. This is critical for many of the authors' conclusions, especially that the phenotypic differences are a result of noise, not levels. The authors must provide a compelling quantitative argument that these levels are the same.

We thank the reviewers for pointing this out. Indeed, as shown in Figure 6A, while unpaired *sens* alleles at both 22A3 and 57F5 locus have identical expression, pairing at the 57F5 locus enhances Sens protein output. Median Sens protein expression for the different *sens* allele pairs is now shown in the new Figure 5—figure supplement 1A, These correspond to the following:

81±2 % increase from 22A3, wild-type → miR-9a mutant

85±2 % increase from 57F5, wild-type → miR-9a mutant

33±1 % increase from wild-type, 22A3 → 57F5

36±2 % increase from miR-9a mutant, 22A3 → 57F5

We show that this minor shift upwards in median Sens levels by 57F5 insertion is uniformly experienced by all cells across the spectrum of Sens expression. This is shown in the new Figure 5—figure supplement 1.

Even though Sens levels only increase by ~30% when expressed from 57F5 compared to 22A3, the bristle error frequency increases 10-fold. In contrast, an ~80% increase in expression due to loss of miR-9a repression does not result in greater bristle pattern errors in either 22A3 or 57F5 animals.

In a different perspective, the miR-9a mutant *sens* paired at 22A3 expresses 36% more protein than the wild type *sens* paired at 57F5. However, its phenotypic error frequency is 3.6% in contrast to 29.1% for wild type *sens* at 57F5 (Figure 7E). The most parsimonious explanation is that noise is the major cause of pattern disorder. We have added text in Results and Discussion to better clarify this argument.

6) The HiC data should be moved to the Supplement.

This has been done. It is now in the new Figure 6—figure supplement 1.

7/8) No experimentation required. Reviewer #1 and reviewer #3 – and then eventually also reviewer #2 – agreed that the most important thing is here is a clarification of the limitations of the assay. Please add sentences to address the ectopic reporter and correlation concerns.

We thank the reviewers for pointing out this necessary qualification. Reviewer 2 asks about whether the alleles are physically paired at 22A3 and 57F5. In our smFISH preprint (Bakker et al., 2020), we successfully identified sites of nascent RNA transcription for the *sens* transgene. Greater than 85% of wing disc cell nuclei contained only one transcription site, not two. This is true whether two transgenic alleles are located at 22A3 or 57F5. When there is only one copy of the transgene at a locus, there is still predominantly one site but it has one-half of the fluorescence intensity. When we probe for *sens* transgene alleles and the endogenous *sens* gene, two transcription sites are detected by smFISH and their locations are not correlated with one another. The same lack of correlation is seen and two sites are observed when a 22A3 and 57F5 transgenic alleles are both present in cell nuclei. We conclude that at both 22A3 and 57F5, homologous alleles are physically co-localized. But 22A3 and 57F5 do not co-localize with one another or the endogenous gene. Unfortunately, the resolution of smFISH using confocal microscopy is not sufficient to see if there are differences in localization that are below the diffraction limit. We have cited and discussed this observation relative to the preprint in the Results.

We have added text in the Discussion that discusses the reviewer’s concerns about limitations in transgene insertion site number and using transgenes in the first place. Clearly more work needs to be done both with *sens* and other genes to see if what we observe with the two sites are more general in relating trans-regulation with expression noise. Note that we performed a cis-trans test for *sens* expression and noise by comparing paired with unpaired alleles. If the cause of the altered expression and noise is due to cis-acting elements, then we would have seen noise and expression changes intrinsic to a single allele. This was not observed. When single alleles were unpaired, the noise and expression was identical to 22A3/22A3.

We would have loved to have tested many insertion sites but given that the experimental work was driven by one person (R.G.) and the experiments and analysis are very laborious, we decided to defer more study of the problem to a future publication. This future work will also include CRISPR modification of the endogenous gene to measure noise.

9) The authors argue that noise drives the phenotypic differences at 22A3 and 57F5 yet the protein levels are the same – this is not a coherent argument: In Figure 7, the authors argue that the levels of Sens protein are the same at 22A3 and 57F5. This is not convincing. The authors should provide a quantitative analysis of the position and expression of Sens in these cells. The position is critical. For example, the central cells could be higher and outer cells could be lower for 22A3 compared to 57F5. This would cause the similarity in quantification of total expression seen in Figure 5C (which is not convincing, as discussed above), yet the spatial differences could cause the phenotype.

We have now worked to convey things more clearly. As discussed earlier in (5), the levels were quantified for different *sens* allele pairs and reported in Figure 5—figure supplement 1A. Median Sens protein levels did not correlate with patterning error frequency.

**Author response image 1. sa2fig1:** Scatterplot of Sens concentration as a function of distance of a cell from the DV boundary. Shown is data from one disc and each datapoint is a single cell.

We examined average Sens expression with respect to distance. The new Figure 7—figure supplement 2 shows line averages of cellular Sens with respect to cell position in individual discs. As expected, average expression peaks closer to the center of the proneural stripes. Since expression from the 57F5 locus is ~30% higher than the 22A3 locus, we see a slightly higher peak for these discs (Figure 7—figure supplement 2B). However, a more striking difference was observed when comparing *sens* alleles with or without miR-9a binding sites (Figure 7—figure supplement 2A). Discs with mutant *sens* alleles showed a much higher expression peak and steeper decay with distance. This is not unexpected since protein levels are ~80% higher in discs with microRNA mutant *sens* alleles. We also compared 22A3 mutant *sens* pairs between panels (A,B) (obtained from different experimental sets) as a control for differences in tissue squishing or stretching due to experimental manipulations (Figure 7—figure supplement 2C).

Patterning error frequency is not different between *sens (+ miR-9a)* and *sens (- miR-9a)* alleles at either locus (Figure 7E). This suggests that if positional differences exist in the level or gradient of Sens, they do not contribute to the observed phenotype. Indeed, this is not surprising since even within a single row of cells equidistant from the Wg secreting margin cells, lateral inhibition generates a salt-and-pepper pattern of S-fated and E-fated cells. This is consistent with our discussion where we surmise that Wg acts as an ‘ON’ switch that sets up the proneural tissue as a whole to express Sens, while Notch-Delta signals help to self-organize individual cells into an ordered S-E pattern.

If Sens expression in 57F5 is perturbed such that a sub-group of cells expresses higher levels relative to 22A3 and another sub-group of cells expresses lower levels relative to 22A3 counterparts, it would necessarily be reflected in the cumulative distribution function (CDF) of Sens at both loci. As shown in the new Figure 5—figure supplement 1B, Sens expressed from 57F5 was consistently higher than Sens expressed from 22A3 throughout the spectrum. There was no intersection of the CDFs as would be expected if sub-groups with higher and lower Sens expression than 22A3 existed in the 57F5 dataset.

In order to better ascertain (than Figure 7A,B) if cells with low or high Sens are differentially distributed in 22A3 vs 57F5 wing discs, we examined the distribution of Sens positive cells according to distance from the DV boundary. The boundary was estimated manually and the shortest distance of each cell centroid from the boundary was calculated. The relationship between Sens level and distance from the wing margin is not trivial. Cells close to the boundary express a broad spectrum of Sens copy numbers, from lowest to highest Sens. As distance increases, the upper limit of copy numbers decreases. Cell positions and protein numbers from a single disc are shown in Author response image 1 to illustrate.

Given that median *sens* expression is higher by ~30% in 57F5 cells, we normalized each cell’s expression by the median Sens level for that genotype. This allowed us to sensitively compare the relative distributions of cellular Sens between 22A3 and 57F5. As shown in Figure 5—figure supplement 1D, the CDFs for both genotypes collapse onto each other. The shaded regions are 95% confidence intervals. This indicates that while total expression is higher in 57F5 cells, the relative distribution of Sens is unaffected i.e. Sens levels uniformly increased over the entire spectrum of expression. This result is consistent with the absence of cell position-specific effects on Sens in 57F5 relative to 22A3. Similar results were obtained when we compared *sens (+ miR-9a)* and *sens (-miR- 9a)* alleles. While the degree of expression difference was much larger, the relative distribution of cells was unaffected (Figure 5—figure supplement 1C, E).

Finally, we wondered if higher error frequencies in 57F5 wings could arise due to greater numbers of cells expressing Sens in 57F5 discs compared to 22A3 (or 22A3/57F5) discs. This was not the case. Disc-wise estimates of mean percentage of Sens-positive cells identified by the automated pipeline are as follows:

22A3/22A3 discs – 34.3 ± 3.7 %

22A3/57F5 discs – 34.9 ± 7.5 %

57F5/57F5 discs – 34.7 ± 3.4 %

We note all of these observations in the Results section.

A bigger issue is the argument that noise drives the phenotype. If the protein levels are identical, it should not matter which allele provides the protein. In other words, if there needs to be 10 units of Sens in a cell, it does not matter if the less noisy 22A3 provide 5 and 5 from each allele whereas the noisier 57F5 provides 4 and 6 from each allele (or 3 and 7, etc.). At the end of the day, the absolute protein quantity should drive the phenotype, not the ratio of protein generated from each allele. The only exception would be if the alleles were different in some way. The only source of difference here would be the tags. If the tags are generating the phenotype, this is also problematic.

Regarding the second point about variable allelic contribution not making a difference to output. Noise is a dynamic process, with stochastic fluctuations in molecule number occurring over time. If we were able to measure Sens molecule number dynamically in a cell, we could characterize these fluctuations and this would be an excellent measure of the noise. However, this technical capability is limited to simple cell systems, primarily in bacteria and yeast. In 2002, Michael Elowitz and Peter Swain devised an elegant alternative method to approximate noise levels without measuring fluctuations over time. This was theoretically laid out in a PNAS paper (2002 vol. 99, 12795) and experimentally demonstrated for bacteria in a Science paper (2002 vol. 297, 1183). Protein number from each allele is stochastically fluctuating over time, and their fluctuations are independent of one another. If one fixes a population of cells and measures protein output from each allele per cell, the limited correlation between allele output for the population closely approximates the stochastic variability if one were to measure noise temporally. Elowitz and Swain’s method has been used many times since then. We are the latest to use it, and have adapted it for *Drosophila* imaginal disc biology.

If one looks at protein level in a population of fixed cells, they are not uniformly expressing identical numbers. Rather, there is a normal distribution of cells with different levels of protein centered around a mean. This is true even for genes that are constitutively expressed and not under regulatory control. The variance in the distribution is because in each cell, protein number stochastically fluctuates, and fluctuations are uncorrelated between cells. Fixation captures one time-point in a highly dynamic and variable process, and misleads one to think that levels do not change in a cell over time. Swain and Elowitz shows that they do and it is a fundamental feature of continuous unregulated gene expression.

Reviewer #3:Major comments:1) In the last paragraph of subsection “Allele Pairing at 57F5 Generates Trans Regulation and Enhanced Noise” and Discussion paragraph two, the authors claim that the noise peak for paired sens alleles at 57F5 results from altered bursting kinetics, specifically an enhanced burst size. Although their modelling supports this claim, there is no direct measurement of RNA bursting kinetics through techniques such as single molecule RNA-FISH. Previous results support the hypothesis of transcriptional burst size modulation (PMID: 24903562) but the authors should also address alternate potential mechanisms (i.e., PMIDs 27153498, 26760529, 26544860), for example the role of alternative splicing (PMIDs: 29986741, 31222776), even if for contrast, as such mechanisms may not be functioning for sens. It should be made clear in the Discussion section that the lack of mRNA quantification is a limitation, and despite existing precedent for burst size modulation, such RNA measurements will be important experiments in future work.

We have cited the references and discussed the issue of limitations in the Discussion, as requested. As noted in responses to reviewers 1 and 2, we now cite a preprint from our lab using smFISH to study *sens* and other genes in the wing. Although we have been unable to perform the most direct experiments to measure burst size or trans regulation, the work does show that RNA data supports *sens* being regulated by burst frequency modulation, as our present protein worked has inferred, These have been noted in the revised manuscript.

Substantive remarks:1) In Figures 4B and 5D the authors have overlaid scatterplots from separate measurements. In regions of high density, there is a loss of information as to where the center of mass lies. The authors should consider using a contour plot or shading to convey density.

We have now done this as requested. We show in those figures hexagonal binning and plot heat maps of data density per bin.

2) Introduction paragraph three the authors list nuclear retention of transcripts as a mechanism for reducing protein noise that may arise from transcriptional bursts. We suggest the authors exercise caution here as the cited papers did not measure protein noise and there is now direct competing evidence indicating that nuclear export amplifies RNA/protein noise in the cytoplasm (PMIDs: 30243562, 30359620). Some might argue that the evidence in this report (increasing protein Fano factor) contradicts the papers cited which claim that nuclear export attenuates noise from transcriptional bursts to minimal Poisson levels.

We have deleted the relevant sentence and citation of retention.

3) In the final paragraph, the authors may want to mention and cite the evidence of other examples where stochastic transcriptional fluctuations appear to have evolved as a mechanism for influencing cell fate (PMIDs: 17379809, 16051143, 28607484).

We have done this as requested.